# GTR-Bench: Evaluating Geo-Temporal Reasoning in Vision-Language Models

**Qinghongbing Xie**[1,2*‡], **Zhaoyuan Xia**[2,3*‡], **Feng Zhu**[2†], **Lijun Gong**[2], **Ziyue Li**[4†], **Rui Zhao**[2],
**Long Zeng**[1 †]
[1]Tsinghua University     [2]SenseTime Research     [3]Peking University
[4]Technical University of Munich, Heilbronn Data Science Center, Munich Data Science Institute
`xqhb23@mails.tsinghua.edu.cn`, `2401210464@stu.pku.edu.cn`,
`zhufeng@sensetime.com`, `zenglong@sz.tsinghua.edu.cn`, `ziyue.li@tum.de`

## Abstract

Recently spatial-temporal intelligence of Visual-Language Models (VLMs) has attracted much attention due to its importance for autonomous driving, embodied AI and general AI. Existing spatial-temporal benchmarks mainly focus on egocentric (first-person) perspective reasoning using images/video contexts, or geographic reasoning with graphical context (e.g., maps), thus fail to assess VLMs' geographic spatial-temporal intelligence that requires integrating both images/video and graphical context, which is crucial for real-world scenarios such as traffic management and emergency response. To address the gaps, we introduce Geo-Temporal Reasoning benchmark (GTR-Bench), a novel challenge for geographic temporal reasoning of moving targets in a large-scale camera network. GTR-Bench is more challenging as it requires multiple perspective switches between maps and videos, joint reasoning across multiple videos with non-overlapping fields of view, and inference over spatial-temporal regions that are unobserved by any video context. Evaluations of more than 10 popular VLMs on GTR-Bench show that even the best proprietary model, Gemini-2.5-Pro (34.9%), significantly lags behind human performance (78.61%) on geo-temporal reasoning. Moreover, our comprehensive analysis on GTR-Bench reveals three major deficiencies of current models for geo-temporal reasoning. (1) VLMs exhibit imbalanced utilization of spatial and temporal context during reasoning. (2) they show weak temporal forecasting ability, leading to poorer performance on temporally focused tasks. (3) they lack the capability to effectively align and integrate map data with multi-view video inputs. We believe GTR-Bench offers valuable insights and opens up new opportunities for research and applications in spatial-temporal intelligence. Benchmark and code will be released at https://github.com/X-Luffy/GTR-Bench.

## 1 Introduction

Spatial intelligence is a fundamental capability that underpins our interaction with the physical world (Feng et al., 2025a). This capability is pivotal for a wide range of applications, including autonomous driving (Cui et al., 2025; Guo et al., 2024) and embodied AI (Chen et al., 2024; Jiang et al., 2025; Huang et al., 2023; Zhang et al., 2025). With recent advances in deep learning, the spatial-temporal intelligence, an extension of spatial intelligence, of Visual-Language Models (VLMs) has emerged as an important research direction. It encompasses the ability to understand spatial attributes such as size and distance, temporal attributes such as time intervals and velocity, and to perform spatial-temporal reasoning about dynamic events in real-world environments.

However, existing benchmarks for the spatial-temporal intelligence of Vision-Language Models (VLMs) have inherent limitations, as shown in Table 1. Current benchmarks for geographic reasoning only focus on static geometric tasks with graphical context such as a subway map (Feng

---

*Equal contribution.
†Corresponding authors.
‡Work done during internships at SenseTime Research.

| Benchmark | Perspective | Geometry vs. Motion | Context Type | #Cam Views |
|---|---|---|---|---|
| ReasonMap(Feng et al., 2025b) | Geographic | Geometry | Graphics | 0 |
| MultiSPA(Xu et al., 2025) | Egocentric | Geometry | Image | 2 |
| STI-BenchLi et al. (2025b) | Egocentric | Geometry & Motion | Video | 1 |
| VSI-BenchYang et al. (2025a) | Egocentric | Geometry & Motion | Video | 1 |
| ViewSpatial-BenchLi et al. (2025a) | Egocentric & Allocentric | Geometry | Image | 2+ |
| ST-VLMKo et al. (2025) | Egocentric | Motion | Video | 1 |
| **GTR-Bench (Ours)** | Geographic | Motion | Graphics & Video | 1 to 3 |

Table 1: **Comparison of GTR-Bench with Previous Benchmarks.** Our GTR-Bench is a novel challenge which can assess VLMs' geographic spatial-temporal intelligence in a camera network with both images/video and graphics context.

et al., 2025b; Chen et al., 2025). In contrast, benchmarks designed for egocentric or allocentric tasks typically involve a single or a few distinct cameras and emphasize scale reasoning between static objects or motion state reasoning of dynamic objects in videos or images (Yang et al., 2025a; Li et al., 2025a; Yang et al., 2025b; Xu et al., 2025; Chen et al., 2025; Feng et al., 2025b; Ko et al., 2025; Li et al., 2025b). These benchmarks therefore fail to evaluate models' ability to perform **geographic spatial-temporal reasoning across large-scale camera networks**, where both graphical context (e.g., maps) and multi-view video observations must be jointly considered. To address these gaps, we propose **Geo-Temporal Reasoning (GTR)**, a novel challenge for geographic temporal reasoning of moving targets in a large-scale camera network. For example, understanding vehicle movement patterns and predicting traffic flow dynamics across a city demands sophisticated reasoning about vehicle appearances, their trajectories, and motion trends across over 10 camera viewpoints. This GTR-Bench is more challenging as it requires multiple perspective switches between maps and videos in a camera network, joint reasoning across multiple videos with non-overlapping fields of view, and inference over spatial-temporal regions that are unobserved by any video.

In this work, we introduce the **Geo-Temporal Reasoning benchmark (GTR-Bench)**, as shown in Figure 1. GTR-Bench features a hierarchical suite of tasks grounded in real-world multi-camera networks. A key innovation of our benchmark is to extend spatial-temporal reasoning tasks to real camera networks within geographic perspective. Grounded in real-world indoor and outdoor scenarios with a multi-camera network, our benchmark utilizes the actual trajectory open-source data of pedestrians and vehicles in urban environments (Tang et al., 2019; Woo et al., 2024) for task construction, including map, object footprint, and trajectory data constructed through annotations. Furthermore, GTR-Bench features a series of tasks for unobserved scenarios that require multi-view joint reasoning across timestamp sequences with non-overlapping camera views. We design a multi-level evaluation framework that separates model capabilities into basic reasoning tasks and combinatorial reasoning tasks, comprising 420 questions derived from 364 videos. The benchmark includes three basic reasoning tasks: *Geo-location*, *Arrival Time-Interval*, and *Motion-State*, which isolate and measure a model's core inferential strengths. To evaluate how models integrate these capabilities, we further design four combinatorial tasks: *Causal Reordering*, *Next Spot Forecasting*, *Trajectory Forecasting*, and *Multi-Target Trajectory Forecasting*.

In our evaluation, models demonstrate markedly poor performance on GTR tasks compared to existing spatial-temporal benchmarks, revealing a critical gap in current VLMs' ability to achieve spatial-temporal intelligence. Even the top-performing proprietary model, Gemini-2.5-Pro, achieved only 34.9% accuracy compared to the human-level performance of 78.61%, while the leading open-source model, InternVL3-38B, reached just 30.76%. This poor performance is further highlighted by two key trends: (1) a significant drop in accuracy from basic to combinatorial tasks, and (2) a notable disparity of performance between outdoor and indoor scenarios. Our in-depth analysis attributes these shortcomings to three fundamental deficiencies in current models. First, VLMs' reasoning is impaired by an imbalanced utilization of context across spatial, temporal, and motion-state dimensions. Second, VLMs are weak in temporal forecasting, which leads to worse performance on spatial-temporal prediction tasks than on spatial reasoning tasks. Third, VLMs lack the proficiency in comprehending and aligning map data with multi-view video inputs from a camera network.

Our main contributions are summarized as follows:

- We propose Geo-Temporal Reasoning, a novel spatial-temporal challenge for geographic temporal reasoning of moving targets in a large-scale camera network. It helps assess VLMs' geographic spatial-temporal intelligence with both visual (images/video) and

graphical (map) contexts, which is critical for applications such as traffic management and emergency response, beyond conventional embodied AI.

- We develop GTR-Bench, a benchmark designed to evaluate geo-temporal reasoning capabilities of VLMs. GTR-Bench is more challenging due to multiple perspective switching between maps and videos, joint reasoning across multiple videos with non-overlapping fields of view, and inference over spatial-temporal regions that are unobserved by any video context. As a result, even the best proprietary model, Gemini-2.5-Pro (34.9%), significantly lags behind human performance (78.61%).

- We provide a comprehensive analysis on GTR-Bench, which reveals three primary deficiencies of current models for geo-temporal reasoning, namely, imbalanced utilization of spatial-temporal context, weakness in temporal forecasting capabilities, and lack of proficiency in comprehending and aligning the map data with multi-view video inputs. We believe analysis on GTR-Bench offers valuable insights and opens up new opportunities for research and applications in spatial-temporal intelligence.

## 2 RELATED WORK

**Geographic Reasoning Benchmark**. Existing benchmarks for geographic reasoning primarily assess models for geometry tasks with graphics context such as a map. ReasoningMap (Feng et al., 2025b) designed a geographic reasoning task for structured and information-rich diagrams like high-resolution transit maps. SpatialLLM (Chen et al., 2025) explores using VLMs to geographic information system data in geometry tasks by transforming multi-modal urban data into structured scene descriptions to prompt pre-trained LLMs. They evaluate the understanding of spatial topology and path planning from a geographic view, yet typically only incorporate evaluation tasks for static geometric targets. In the GTR-Bench, we combine geographic data to reason about dynamic objects. It will bring a new cognitive perspective to the existing spatial-temporal benchmarks.

**Spatial-Temporal Reasoning Benchmark**. Existing benchmarks for spatial-temporal reasoning assess models on video or multi-image tasks from a single or a few camera views. ST-VLM (Ko et al., 2025) constructs a dataset and benchmark for kinematic instruction tuning (STKit/STKit-Bench) to propose and validate ST-VLM, which demonstrates outstanding performance in object dynamics analysis. STI-Bench (Li et al., 2025b) evaluates the capabilities of VLMs on real-world spatio-temporal understanding tasks such as pose, displacement, and motion. They focus on event sequencing and state inference within a single video from an egocentric view. ViewSpatial-Bench (Li et al., 2025a) addresses the core problem of VLMs' inadequate spatial reasoning when switching from egocentric to allocentric perspectives. MMSI-Bench (Yang et al., 2025b) is built upon spatial reasoning tasks that span the positions, attributes, and motions with a multi-step reasoning split that chains them into long-horizon questions. Multi-MultiSPA (Xu et al., 2025) is designed for multi-frame spatial reasoning covering depth and visual correspondence perception, camera and object movement perception, and object size perception. They focused on the challenges of multi-view tasks and attempted to evaluate models to perform spatial-temporal reasoning under few adjacent yet distinct views as an egocentric or allocentric observer. Although these spatial-temporal works have some similarities with GTR, GTR notably combines graphic map and video as context and proposes a geographic temporal reasoning task of moving targets with multi-view joint reasoning with little view overlap for unobserved scenes.

**Geo-Temporal Task in Multi-Camera Systems**. Geo-temporal task represents a type of spatial-temporal tasks that covers large-scale camera networks across multiple regions/districts, necessitating comprehension of geographic relationships and cross-regional connections. Multi-Target Multi-Camera Tracking (MTMCT) can be considered as a form of geo-temporal task. Existing benchmarks for MTMCT include the CityFlow dataset (Tang et al., 2019), which provides vehicle trajectories in a large urban area, and MTMMC (Woo et al., 2024), which offers pedestrian trajectories in indoor environments. Complementing these datasets, various methods have been developed for multiple camera Re-ID. These include two-step matching approaches using semantic parsing and spatial-temporal attention (He et al., 2020), the integration of language models with graph neural networks (Nguyen et al., 2024), and noise-robust trajectory recovery frameworks designed to address Re-ID clustering errors (Li et al., 2025c). However, existing researches still rely primarily on visual features of objects, which remain within the computer vision domain with limited generalizability.

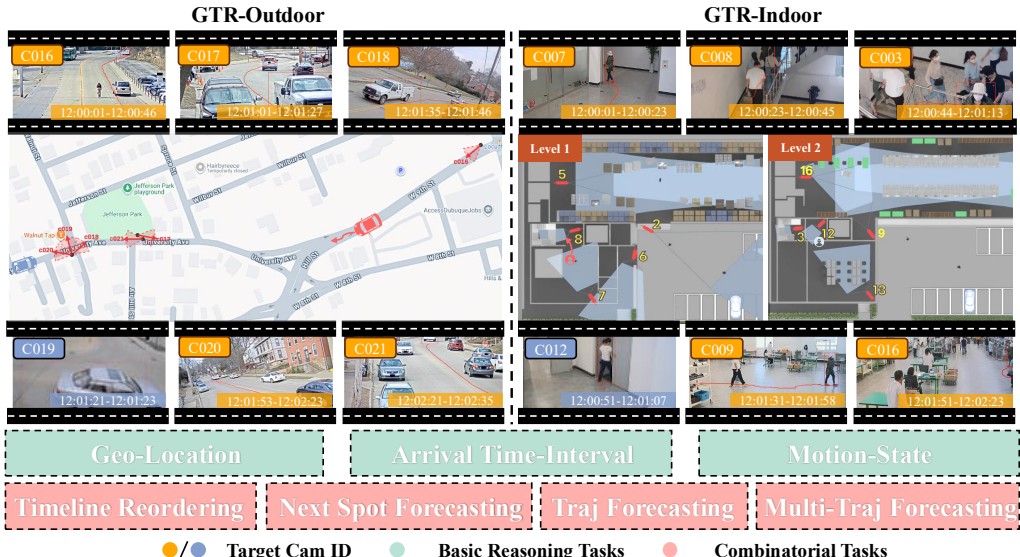

Figure 1: **Overview of Geo-temporal Reasoning and GTR-Bench.** Given a graphical map and multiple video clips from non-overlapping cameras, geo-temporal reasoning infers motion state of moving targets in a large-scale camera network. GTR-Bench comprises 3 basic reasoning tasks, including Geo-location, Arrival Time-Interval, and Motion-State, and 4 combinatorial tasks including Causal Reordering, Next Spot Forecasting, Trajectory Forecasting, and Multi-Target Trajectory Forecasting. GTR-Bench covers both outdoor (vehicles) and indoor (pedestrians) scenarios.

Given the extensive exploration of VLMs in spatial-temporal intelligence applications, we extend the geo-temporal task to a novel challenge, Geo-temporal Reasoning(GTR), which leverages the reasoning capabilities of VLMs to fully utilize geographic and temporal context in solving more challenge.

# 3   GEO-TEMPORAL REASONING

From a cognitive science perspective, spatial-temporal intelligence can be bifurcated into two categories: first-person (egocentric) and third-person (allocentric) intelligence (Burgess, 2006). However, if we move away from the human cognitive perspective and perceive the real world from a broader perspective, the geographic perspective can provide VLMs with an omniscient understanding for dynamic objects.

When we extend the context of the problem to geographic spatial-temporal intelligence, the issue becomes more challenging and valuable, requiring the resolution of multi-perspective changes and transformations of coordinate systems. A typical case involves inferring a target's geo-location, temporal sequence, and motion state in a camera network comprising more than 10 viewpoints covering an urban scene. We posit that this represents a crucial challenge of spatial-temporal intelligence, which we define as Geo-Temporal Reasoning (GTR), as illustrated in Figure 1.

With the expansion of contextual inputs from the GTR challenge, novel questions have arisen for existing spatial-temporal intelligence. The introduction of a graphic map requires models to reason between the map and cameras, handling multiple perspective changes. This contrasts sharply with egocentric tasks, which rely on a continuous-time sequence inferred from the sequential order of image frames (Xiong et al., 2024; Su et al., 2024; Bazaga et al., 2025). Furthermore, GTR challenges VLMs to perform multi-view joint reasoning about unobserved phenomena with little view overlap, such as inferring a target's path between camera views. This demands superior inferential capabilities and context integration. In contrast, egocentric tasks typically involve continuous and unified spatial-temporal observations, allowing models to reason about directly observed events (Ko et al., 2025; Li et al., 2025b).

| Task Name | Task Definition | Metric |
|---|---|---|
| **Basic Tasks** | | |
| Geo-Location (GL) | Given the starting and ending locations of a target, infer the intermediate locations the target passes through. *Example: Based on the provided [start video] and [end video], infer which camera the target passed through between the start point (C016, 12:00:10-12:00:22) and end point (C018, 12:00:37-12:00:42). Answer: C. C017* | MCQ Acc |
| Arrival Time-Interval (ATI) | Given the starting point, ending point, and intermediate location, infer the time interval of the target's arrival at a specific intermediate location. *Example: Based on the provided [start video] (C018, 12:00:37-12:00:42) and [end video] (C020, 12:00:43-12:00:50),and knowing the target passed through camera C019, infer when the target arrived at the intermediate camera. Answer: A. 12:00:43.279-12:00:43.579* | MCQ Acc |
| Motion-State (MS) | Given the starting point, ending point, and intermediate location, infer the plausible motion state of the target at intermediate locations. *Example: Based on the provided [start video] (C016, 12:00:10-12:00:22) and [end video] (C018, 12:00:37-12:00:42),and the intermediate camera c017 infer the target's motion state during the intermediate time period. Answer: B. the target travels west at a speed of 10.0 m/s for 11.0 seconds, covering a distance of 109.6 meters.* | MCQ Acc |
| **Combinatorial Tasks** | | |
| Causal Reordering (CR) | Given a set of unordered video clips from different cameras and a map, determine the correct chronological sequence of cameras the target passed through. *Example: Based on the provided [local map] and [videos](C019,C021,C020), analyze the target's activity trajectory. Please infer the correct order in which the target passed through these cameras. Answer: D. C019 → C020 → C021* | MCQ Acc |
| Next Spot Forecasting (NSF) | Given the target's last observed appearance in a single camera video and a map, predict the most probable next camera location and the corresponding time interval of appearance. *Example: Based on the provided [local map] and [video] (C16, 12:00:10-12:00:22), which camera from the following list will likely capture the target next? You need to select one option as the answer and infer a time range. Answer: A. C020 12:00:43.905-12:00:50.505* | ST-IoU |
| Trajectory Forecasting (TF) | Building upon multiple historical observations across several cameras, predict the target's complete future trajectory by forecasting the sequence of cameras it will pass through. *Example: Based on the provided [local map] and [videos] (C017, 12:01:01-12:01:27, C018, 12:00:37-12:00:42), predict the next two cameras that the target will likely pass through. You need to select a correct sequence of options and simultaneously infer a corresponding sequence of time ranges. Answer: A. C019 12:00:43.279-12:00:43.579 → D. C020 12:00:43.905-12:00:50.505* | ST-IoU |
| Multi-Target Trajectory Forecasting (MTTF) | This extends single-target prediction by requiring the model to forecast the future meeting point (location and time) of two distinct targets. *Example: Based on the provided [local map] and [videos] (C018, 12:00:37-12:00:42, C019, 12:01:21-12:01:23) showing the movement trajectories of two [Target], predict where and when these [Target] will most likely meet. Answer: B. C018 12:00:37.755-12:00:42.855* | ST-IoU |

Table 2: **Detailed design and examples of tasks in GTR-Bench.** GTR-Bench is divided into basic and combinatorial levels, evaluated by either Multiple-Choice Question Accuracy (MCQ Acc) or Spatial-Temporal Intersection over Union (ST-IoU).

# 4 GTR-BENCH

## 4.1 OVERVIEW OF GTR-BENCH

In this work, we constructed the Geo-Temporal Reasoning benchmark (GTR-Bench) to systematically evaluate VLMs on geo-temporal reasoning challenge, as in Table 2. (1) The first level comprises three basic reasoning tasks designed to probe fundamental abilities. **Geo-Location** assesses a model's comprehension of spatial topology and path planning within multi-camera networks. **Arrival Time-Interval** evaluates the ability to model temporal sequences and predict event timing across different camera views. **Motion-State** examines the understanding of target's behavior and the influence of scene semantics on motion state. (2) The second level features four combinatorial tasks that demand the integration of three basic reasoning skills. **Causal Reordering** requires the synthesis of spatial and temporal understanding to reconstruct a coherent event sequence. **Next Spot Forecasting** integrates all three basic skills to predict a target's subsequent location and time of appearance in the future. **Trajectory Forecasting** extends this by requiring long-horizon prediction, testing the model's ability to understand sustained motion patterns. **Multi-Target Trajectory Forecasting** testing joint reasoning and the capacity to manage multiple spatial-temporal paths.

For each task in the GTR-Bench, we have designed specific problem instances and corresponding evaluation metrics, as detailed in Table 2. Each question is multi-modal, incorporating a map with

one or more video clips, and features a rich diversity of target objects and spatial-temporal contexts. Overall, the benchmark comprises 420 unique questions derived from 364 distinct video clips with an average duration of 10.64 seconds. As shown in Table 3, these questions are meticulously balanced across the seven reasoning tasks, with 60 questions per task. More examples and details are available in Appendix G.

The benchmark is equally divided into two distinct real-world scenarios, with 210 questions for each. The first is an outdoor urban environment from the CityFlow dataset (Tang et al., 2019) focused on vehicles, while the second is an indoor, multi-level setting from the MTMMC dataset (Woo et al., 2024) for pedestrians. As detailed in Table 3, these environments present vastly different scales of geo-temporal complexity. The outdoor network spans a city block with 31 cameras and an average inter-camera distance of 984.77 meters. In contrast, the indoor network provides building-level coverage with 16 cameras and an

| Metric | Overall Value |
|---|---|
| Total Questions | 420 |
| Unique Videos | 364 |
| Avg. Images per Question | 7.0 |
| Avg. Question Length | 185.1 |
| Avg. Choice Length | 132.4 |

| Sub-Dataset Details | GTR-Outdoor | GTR-Indoor |
|---|---|---|
| Target Objects | Vehicles | Pedestrian |
| Number of Questions | $7 \times 30$ | $7 \times 30$ |
| Number of Cameras | 31 | 16 |
| Avg. Distance (m) | 984.77 | 30.59 |
| Max. Distance (m) | 2389.42 | 60.93 |

Table 3: Data Statistics

average distance of only 30.59 meters. This significant disparity, with the outdoor scenario being approximately 32 times larger in spatial scale, establishes a testbed for evaluating the spatial-temporal intelligence of VLMs.

The design of GTR-Bench prioritizes diverse spatial-temporal complexity over static backgrounds, grounding the evaluation of VLMs in dynamic cues rather than environmental textures. We categorize tasks into three complexity tiers of Geo-temporal complexity—Long, Medium, and Short—based on physically-grounded thresholds for trajectory length ($track_d$) and duration ($track_t$) across both outdoor and indoor settings as Fig. 2. This categorization ensures a balanced and comprehensive distribution of tasks as Fig. 3, enabling a rigorous assessment of geo-temporal reasoning across varying spatial and temporal scales.

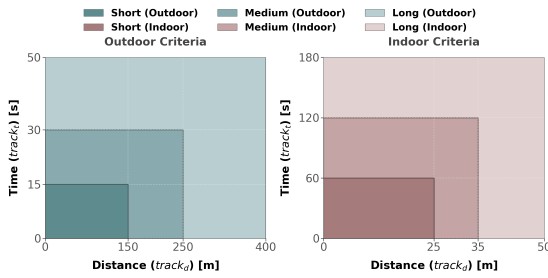

Figure 2: Complexity definition for outdoor and indoor: outdoor is generally quicker in time and longer in distance due to driving then indoor.

Figure 3: Complexity Distribution: three levels of complexity are quite balanced for both outdoor and indoor.

## 4.2 METRIC DESIGN

We employ two primary metrics for evaluation: standard accuracy for multiple-choice questions (MCQs) and a novel Spatial-Temporal Intersection over Union (ST-IoU) for predictive tasks, as outlined in Table 2. For the basic and CR tasks, which are formatted as MCQs, we report standard accuracy. In contrast, the forcasting tasks (NSF, TF, and MTTF) require the model to generate both a Camera ID (e.g., *C.C017*) and a corresponding time interval (e.g., *12:00:37-12:00:42*). To evaluate spatial-temporal results, we introduce ST-IoU, a composite metric designed to assess spatial-temporal performance. Specifically, it combines spatial accuracy—verifying the correctness of the predicted Camera ID—with Temporal IoU, which measures the overlap between the predicted and ground-truth time intervals. For a given prediction $i$, the ST-IoU is calculated as follows:

$$\text{ST-IoU} = \frac{1}{N} \sum_{i=1}^{N} \mathbb{I}(C_{p_i} = C_{gt_i}) \times \frac{|T_{p_i} \cap T_{gt_i}|}{|T_{p_i} \cup T_{gt_i}|} \qquad (1)$$

where $N$ is the total number of predictions, $\mathbb{I}(\cdot)$ is the indicator function which equals 1 if the predicted camera $C_{p_i}$ matches the ground truth camera $C_{gt_i}$ and 0 otherwise. $T_{p_i}$ and $T_{gt_i}$ represent the predicted and ground-truth time intervals and the fraction calculates their temporal IoU.

## 4.3 BENCHMARK CONSTRUCTION PIPELINE

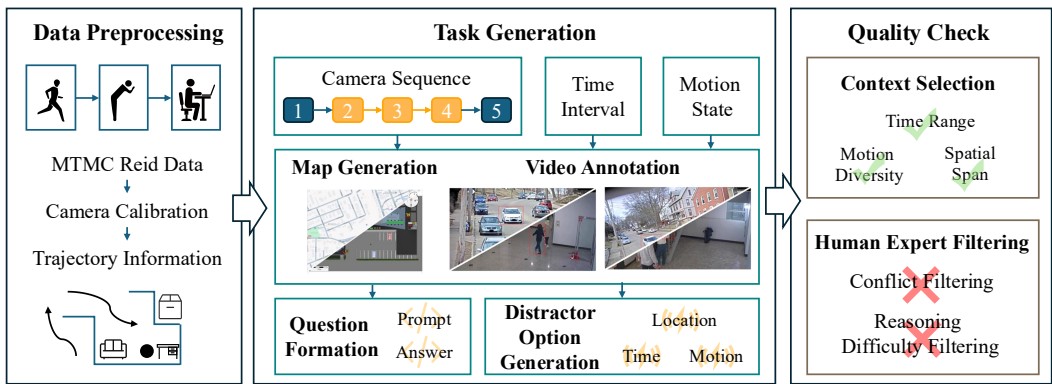

Figure 4: Benchmark construction pipeline

We designed an automated benchmark construction pipeline that transforms raw video data into standardized questions across seven tasks to accommodate the varying temporal, geographic, and formatting requirements of different tasks, as shown in Figure 4.

**Data Preprocessing**. Our data preprocessing pipeline transforms raw video from two distinct datasets, CityFlow (outdoor vehicles)(Tang et al., 2019) and MTMMC (indoor pedestrians)(Woo et al., 2024), into a structured format suitable for geo-temporal reasoning. The process begins by segmenting video clips for individual target instances based on bounding box annotations. Next, we perform camera calibration by computing a homography matrix from manually annotated correspondence points, which establishes a precise projective transformation from the 2D image plane to a real-world map. Using this matrix, we transform target trajectories by projecting their coordinates onto the map, scaling them to real-world metrics, and algorithmically deriving key motion parameters like velocity and direction. The resulting trajectory data then undergoes a rigorous cleaning, filtering, and validation process to ensure consistency. Finally, we use a large language model to synthesize this quantitative data into a qualitative *Motion Summary*, providing both numerical and narrative insights for each trajectory. For more details, refer to Appendix A.1.

**Task Construction**. We employ a systematic pipeline to create standardized questions using task-specific templates. The process consists of four main steps: **(1) Trajectory Selection**: We randomly sample valid trajectory segments, complete with temporal and motion state data, from our preprocessed dataset. **(2) Information Integration**: The selected trajectory, along with corresponding map information and video data, is integrated into a standardized template. **(3) Question and Answer Formulation**: We construct a task question aligned with specific reasoning requirements and establish the ground-truth answer based on the actual trajectory and map data. **(4) Distractor Generation**: To ensure a rigorous evaluation, we generate plausible yet incorrect distractor options using a sophisticated, scenario-specific strategy. This includes sourcing from architecturally distinct indoor areas, algorithmically creating synthetic outdoor cameras, and randomizing camera IDs to compel models to perform genuine geo-temporal reasoning rather than relying on superficial heuristics. For more details, refer to Appendix A.3.

**Quality Check**. The benchmark has undergone a complete two-stage manual selection process to ensure the validity of the evaluation. In the first stage, we first manually select the context of each question to ensure the diversity of spatial spans and temporal durations of the questions. Meanwhile, we remove questions with large trajectory errors to bring the benchmark more in line with the laws of spatial-temporal reasoning. In the second stage, human experts select the answers of each question and select 30 questions per task covering reasonable difficulty levels. This ensures the validity of the benchmark and provides diversity in evaluation difficulty.

## 5 EVALUATION ON BENCHMARK

### 5.1 EVALUATION SETUP

**Evaluation Setup.** We evaluate 13 state-of-the-art models across our benchmark. For proprietary models (PM), we select Claude-3.7-Sonnet (Anthropic, 2025a), Claude-4-Sonnet (Anthropic, 2025b), GPT-4o (Openai, 2024), GPT-5 (Openai, 2025), and Gemini-2.5-Pro (Deepmind, 2025). For open-source models (OM), we include the InternVL3-2B/8B/38B series (Zhu et al., 2025), Qwen2-VL-2B/7B series (Wang et al., 2024), Qwen2.5-VL-2B/7B/32B series (Bai et al., 2025), and GLM-4.1V-9B-Thinking (Hong et al., 2025). Proprietary models are accessed via their respective official APIs, while open-source models are deployed using LMDeploy on 8 NVIDIA V100 GPUs. To accommodate model input constraints, we uniformly sample the videos, ensuring the total number of frames from multiple videos remains within 20. We set the *temperature* to 0.1 and *max_new_token* to 16,384, enabling the models to perform sufficient and stable chain-of-thought reasoning. Additionally, a traditional ReID-based baseline is implemented to provide a performance reference, detailed in Appendix C.1.

### 5.2 MAIN RESULTS

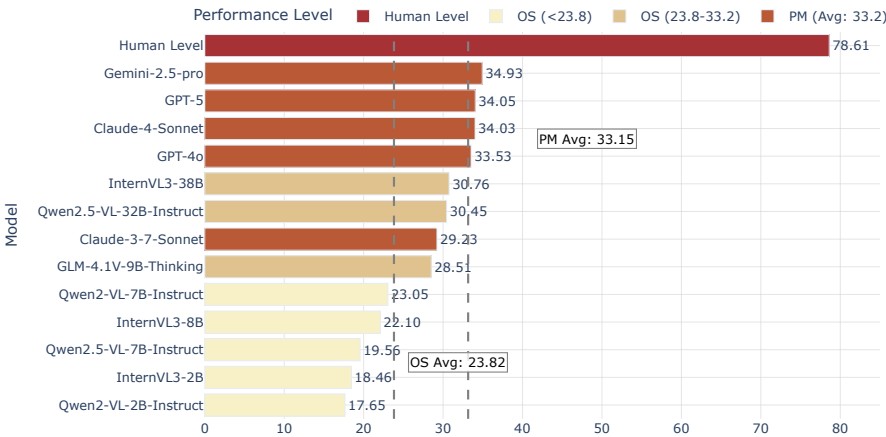

Figure 5: Overview Comparison of human performance and 13 VLMs on GTR-Bench. The best model (Gemini-2.5-Pro, 34.93%) remains far below human performance (78.61%). Dashed lines indicate average performance: OS Avg (23.82) < PM Avg (33.15) ≪ Human (78.61%).

**Overview of Results.** A critical performance gap is notable, with even the best proprietary model, Gemini-2.5-Pro, achieving a score of only 34.9, compared to the human-level performance of 78.61. Table 4 presents a comprehensive evaluation of 12 Visual-Language Models on our GTR-Bench, spanning two distinct scenarios (GTR-Outdoor and GTR-Indoor) and seven reasoning tasks. The proprietary models, particularly Gemini-2.5-Pro, GPT-5 and Claude-4-Sonnet, demonstrate better performance, achieving the highest average scores of 34.93, 34.05 and 34.03, respectively. This indicates that current leading proprietary models possess more robust geo-temporal reasoning capabilities. However, the performance gap is narrowing, with top-tier open-source models like InternVL3-38B, Qwen2.5-VL-32B, and GLM-4.1V-9B-Thinking showing competitive results with scores of 30.76, 30.45, and 28.51 respectively. The overall scores reveal the

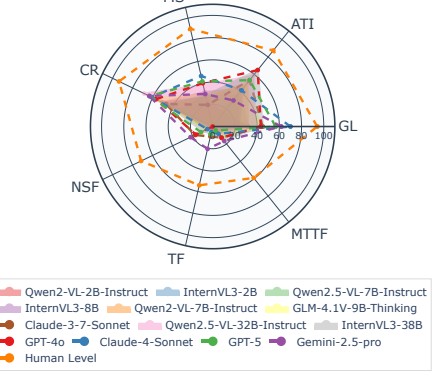

Figure 6: Task performance of GTR-Bench

challenge of GTR-Bench and highlight significant room for improvement in the spatial-temporal intelligence of VLMs.

| Methods | Rank | GTR-Outdoor | | | | | | | GTR-Indoor | | | | | | | Average |
|---|---|---|---|---|---|---|---|---|---|---|---|---|---|---|---|---|
| | | GL | ATI | MS | CR | NSF | TF | MTTF | GL | ATI | MS | CR | NSF | TF | MTTF | |
| *Proprietary Models (API)* | | | | | | | | | | | | | | | | |
| Claude-3-7-Sonnet | 7 | 53.33 | 66.67 | 26.67 | 46.67 | **25.75** | 8.90 | 21.97 | 36.67 | 40.00 | 13.33 | 46.67 | 9.48 | 9.51 | 3.56 | 29.23 |
| GPT-4o | 4 | 56.67 | **76.67** | 40.00 | 63.33 | 20.53 | 0.00 | **23.10** | 30.00 | 53.33 | 40.00 | 50.00 | 13.00 | 0.00 | 2.79 | 33.53 |
| Claude-4-Sonnet | 3 | **73.33** | 50.00 | **50.00** | 63.33 | 8.05 | 6.18 | 16.94 | **66.67** | 33.33 | 43.33 | 58.62 | 2.60 | 4.01 | 0.00 | 34.03 |
| GPT-5 | 2 | 53.33 | **76.67** | 40.00 | 40.00 | 12.04 | 12.12 | 7.34 | 60.00 | 30.00 | **43.33** | **86.21** | 11.34 | 2.55 | 1.75 | 34.05 |
| Gemini-2.5-Pro | 1 | 60.00 | 46.67 | 33.33 | 56.67 | 19.13 | **13.16** | 19.18 | 63.33 | 13.33 | 26.67 | 70.00 | **25.11** | **28.09** | **14.37** | **34.93** |
| *Open-source Models* | | | | | | | | | | | | | | | | |
| Qwen2-VL-2B-Instruct | 13 | 33.33 | 16.67 | 20.00 | 56.67 | 0.00 | 0.28 | 0.00 | 30.00 | 13.33 | 33.33 | 43.33 | 0.00 | 0.21 | 0.00 | 17.65 |
| InternVL3-2B | 12 | 33.33 | 46.67 | 23.33 | 36.67 | 6.15 | 0.00 | 9.95 | 13.33 | 23.33 | 33.33 | 30.00 | 0.65 | 0.08 | 1.62 | 18.46 |
| Qwen2.5-VL-7B-Instruct | 11 | 23.33 | 46.67 | 3.33 | 60.00 | 0.00 | 0.00 | 0.51 | 40.00 | 30.00 | 6.67 | 63.33 | 0.00 | 0.00 | 0.00 | 19.56 |
| InternVL3-8B | 10 | 26.67 | 60.00 | 33.33 | 50.00 | 0.00 | 4.79 | 5.42 | 20.00 | 26.67 | 30.00 | 50.00 | 0.00 | 0.79 | 1.67 | 22.10 |
| Qwen2-VL-7B-Instruct | 9 | 43.33 | 60.00 | 16.67 | 50.00 | 5.78 | 0.00 | 10.01 | 20.00 | 40.00 | 36.67 | 36.67 | 3.62 | 0.00 | 0.00 | 23.05 |
| GLM-4.1V-9B-Thinking | 8 | 60.00 | 57.14 | 25.00 | 62.07 | 10.29 | 0.00 | 25.38 | 26.67 | 38.46 | 34.48 | 55.17 | 2.87 | 0.00 | 1.67 | 28.51 |
| Qwen2.5-VL-32B-Instruct | 6 | 43.33 | 60.00 | 33.33 | **66.67** | 0.65 | 0.00 | 15.72 | 33.33 | **56.67** | **43.33** | 70.00 | 3.33 | 0.00 | 0.00 | 30.45 |
| InternVL3-38B | 5 | 40.00 | 73.33 | 30.00 | 53.33 | 8.27 | 0.00 | 20.58 | 50.00 | **56.67** | 26.67 | 37.93 | 11.10 | 4.37 | 10.24 | 30.76 |
| Human Level | - | 90.00 | 84.25 | 90.91 | 89.75 | 68.31 | 51.24 | 55.83 | 98.20 | 90.78 | 89.45 | 97.35 | 74.64 | 57.36 | 62.46 | 78.61 |
| ReID Baseline | - | 63.33 | 23.33 | - | - | 66.67 | 38.33 | 66.67 | 46.67 | 16.67 | - | - | 43.33 | 40.00 | 52.17 | 45.72 |

Table 4: **GTR-Bench Results.** Our evaluation based on more than 10 popular VLMs reveals a critical performance gap. **Bold** = best result; Underline = second best result.

**Outdoor vs Indoor**. A comparative analysis between the GTR-Outdoor and GTR-Indoor scenarios reveals distinct performance patterns. As shown in Table 4, most models demonstrate better performance in outdoor settings. For instance, Claude-3.7-Sonnet achieves outdoor scores of 25.75, 8.90, and 21.97 for NSF, TF, and MTTF tasks respectively, consistently outperforming its indoor scores of 9.48, 9.51, and 3.56. This aligns with expectations that outdoor environments provide clearer spatial cues and more regular motion patterns. However, Gemini-2.5-Pro exhibits counterintuitive behavior, achieving higher indoor scores of 25.11, 28.09, and 14.37 compared to its outdoor scores of 19.13, 13.16, and 19.18. This pattern, demonstrated by the top-performing model, suggests that advanced models may better utilize their sophisticated reasoning capabilities when facing more complex indoor scenarios that demand deeper multi-dimensional reasoning.

## 5.3 RESULT ANALYSIS

**Context Utilization Analysis**. We designed a prompt to analyze each model's reliance on spatial, temporal, and motion state context, detailed in Appendix D, which instructs an LLM to assess its own reasoning output for each question. The model assigns a binary score for its utilization of each context type: 0 for no utilization and 1 for

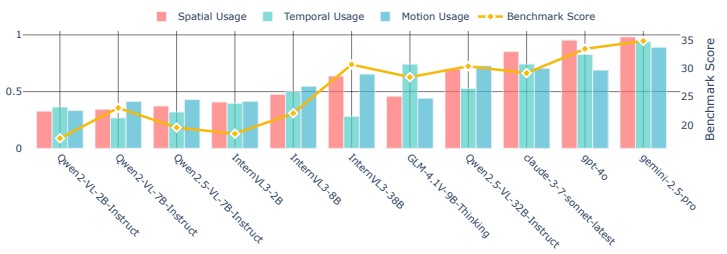

Figure 7: **Context Utilization Analysis**. VLMs' reasoning is impaired by an imbalanced utilization of spatial-temporal context.

appropriate utilization. This method allows us to quantify each model's dependency on different context types and reveal its reasoning patterns. As shown in Figure 7, our analysis reveals that leading proprietary models like Gemini-2.5-Pro demonstrate balanced utilization across all context types, which contributes to their superior performance. In contrast, open-source models often exhibit imbalanced patterns. For instance, InternVL3-38B shows strong spatial and motion understanding but weaker temporal reasoning. This imbalance creates reasoning blind spots that limit performance on comprehensive geo-temporal tasks.

**Spatial-temporal Reasoning Analysis**. To investigate the performance gap between spatial and spatial-temporal reasoning, we conducted a detailed analysis as detailed in Table 5. Models achieve higher MCQ accuracy evaluating spatial location understanding compared to ST-IoU scores requiring joint spatial-temporal reasoning. GPT-4o achieves 53.33 MCQ accuracy on outdoor NSF but only 20.53 ST-IoU, a 32.80 point gap. Gemini-2.5-Pro shows 45.45 MCQ accuracy versus 13.16 ST-IoU on outdoor TF, indicating a 32.29 point difference. This pattern persists across all models, with GLM-4.1V-9B-Thinking demonstrating 76.67 MCQ accuracy on outdoor MTTF but only 25.38 ST-IoU, a 51.29 point difference. These consistent gaps highlight that while models identify spatial

| Methods | Rank | GTR-Outdoor | | | | | | GTR-Indoor | | | | | |
|---|---|---|---|---|---|---|---|---|---|---|---|---|---|
| | | NSF | | TF | | MTTF | | NSF | | TF | | MTTF | |
| | | MCQ Acc | ST-IoU | MCQ Acc | ST-IoU | MCQ Acc | ST-IoU | MCQ Acc | ST-IoU | MCQ Acc | ST-IoU | MCQ Acc | ST-IoU |
| *Proprietary Models (API)* | | | | | | | | | | | | | |
| Claude-3-7-Sonnet | 7 | 36.67 | 25.75 | 41.67 | 8.90 | 73.33 | 21.97 | 23.33 | 9.48 | 18.33 | 9.51 | 6.67 | 3.56 |
| GPT-4o | 4 | 53.33 | 20.53 | 41.67 | 0.00 | 76.67 | 23.10 | 30.00 | 13.00 | 38.33 | 0.00 | 13.33 | 2.79 |
| Claude-4-Sonnet | 3 | 36.67 | 8.05 | 45.00 | 6.18 | 70.00 | 16.94 | 30.00 | 2.60 | 21.67 | 4.01 | 16.67 | 0.00 |
| GPT-5 | 2 | 73.33 | 12.04 | 58.33 | 12.12 | 83.33 | 7.34 | 50.00 | 11.34 | 38.33 | 2.55 | 26.67 | 1.75 |
| Gemini-2.5-Pro | 1 | 38.46 | 19.13 | 45.45 | 13.16 | 51.72 | 19.18 | 43.33 | 25.11 | 50.00 | 28.09 | 36.67 | 14.37 |
| *Open-source Models* | | | | | | | | | | | | | |
| Qwen2-VL-2B-Instruct | 13 | 36.67 | 0.00 | 30.00 | 0.28 | 43.33 | 0.00 | 3.33 | 0.00 | 23.33 | 0.21 | 10.00 | 0.00 |
| InternVL3-2B | 12 | 33.33 | 6.15 | 18.33 | 0.00 | 27.59 | 9.95 | 6.67 | 0.65 | 8.33 | 0.08 | 3.57 | 1.62 |
| Qwen2.5-VL-7B-Instruct | 11 | 43.33 | 0.00 | 28.33 | 0.51 | 16.67 | 0.51 | 16.67 | 0.00 | 20.00 | 0.00 | 20.00 | 0.00 |
| InternVL3-8B | 10 | 40.00 | 0.00 | 23.33 | 4.79 | 36.67 | 5.42 | 13.33 | 0.00 | 15.00 | 0.79 | 6.67 | 1.67 |
| Qwen2-VL-7B-Instruct | 9 | 43.33 | 5.78 | 26.67 | 0.00 | 51.72 | 10.01 | 26.67 | 3.62 | 18.33 | 0.00 | 6.67 | 0.00 |
| GLM-4.1V-9B-Thinking | 8 | 40.00 | 10.29 | 30.00 | 0.00 | 76.67 | 25.38 | 10.34 | 2.87 | 0.00 | 0.00 | 6.67 | 1.67 |
| Qwen2.5-VL-32B-Instruct | 6 | 40.00 | 0.65 | 26.92 | 0.00 | 50.00 | 15.72 | 10.00 | 3.33 | 26.00 | 0.00 | 6.67 | 0.00 |
| InternVL3-38B | 5 | 23.33 | 8.27 | 23.33 | 8.20 | 50.00 | 20.58 | 23.33 | 11.10 | 16.67 | 4.37 | 16.67 | 10.24 |

Table 5: **Spatial-temporal Reasoning Analysis.** VLMs' weakness in temporal forecasting causes worse performance on spatial-temporal prediction tasks (ST-IoU) than spatial reasoning tasks (MCQ Acc).

locations adequately, they struggle when temporal constraints are integrated into reasoning, suggesting VLMs lack temporal reasoning mechanisms for comprehensive geo-temporal understanding.

**Failure Case Study**. We conduct a comprehensive error analysis on representative models, including Gemini-2.5-Pro, GPT-4o, and InternVL3-32B. Our analysis reveals novel error patterns emerging from the complex interplay of multiple discontinuous perspectives and explicit temporal reasoning, as illustrated in Figure 8. In Example 1, model fails to understand wall obstructions and calculates routes using direct linear distance, resulting in Topology Error. Model correctly identifies video direction but encounters failures during world direction transformation, leading to Motion State Error. In Example 2, model fails to properly map camera FoV to the spatial layout, erroneously equating location with viewpoint and causing FoV Alignment Error. Model ignores distance and speed constraints, incorrectly estimating traversal time and resulting in Time Interval Error. Detailed definitions and comprehensive analysis are provided in Appendix E.

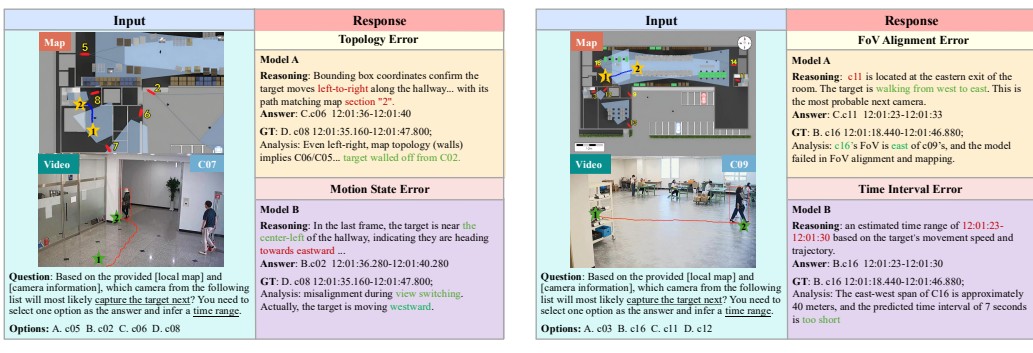

(a) Example 1        (b) Example 2

Figure 8: **Failure Case Study.** VLMs lack the proficiency to comprehend or align the map data with multi-view video inputs. (a) Example 1: Topology Error and Motion State Error. (b) Example 2: FoV Alignment Error and Time Interval Error.

## 6 CONCLUSION

In this work, we introduced the Geo-Temporal Reasoning benchmark, a novel challenge for evaluating the spatial-temporal intelligence of Visual-Language Models through geographic temporal reasoning tasks of moving targets in real-world multi-camera networks with large perspective changes. Our comprehensive evaluation reveals significant limitations in current models. The best-performing proprietary model, Gemini-2.5-Pro, achieves only 34.9% accuracy, highlighting a critical gap behind human performance (78.61%). Our analysis suggests these shortcomings stem from imbalanced utilization of spatial-temporal context, weakness in temporal forecasting capabilities, and lack of proficiency in comprehending and aligning the map data with multi-view video inputs. We believe GTR-Bench offers valuable insights and opens up new opportunities for research and applications in spatial-temporal intelligence.

ANNOUNCEMENT

In an effort to clearly convey the innovative ideas presented in this paper, we utilized a Large Language Model to polish and optimize the manuscript's expression, thereby making the content more accessible to the reader.

ACKNOWLEDGMENT

This work is supported by Shenzhen Science and Technology Program (Grant No. CJGJZD20240729141702003).

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

# A MORE DETAILS OF THE BENCHMARK CONSTRUCTION PIPELINE

## A.1 DATA PREPROCESSING DETAILS

**Data Collection**. To obtain richer scenarios and diverse moving targets, we collected data from two completely distinct scenarios, **Outdoor** and **Indoor**, to construct the GTR-Bench. For outdoor data, we used the CityFlow dataset, which is an urban Multi-Target Multi-Camera (MTMC) video dataset containing 345 target id of cars and 40 cameras.(Tang et al., 2019) For indoor data, we used the MTMMC dataset, an indoor MTMC video dataset that includes 3669 target id of people and 32 cameras.(Woo et al., 2024) These datasets provide the Re-identification (ReID) information of each target in the video as well as the temporal information of bounding boxes (bbox). Starting with the raw video sequences and associated detection annotations, we segment and crop frames based on target bounding boxes. This procedure isolates individual target instances, forming a discretized visual dataset that serves as the foundation for subsequent analysis.

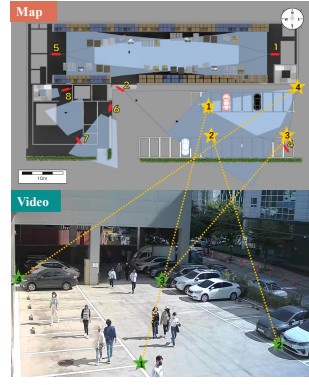

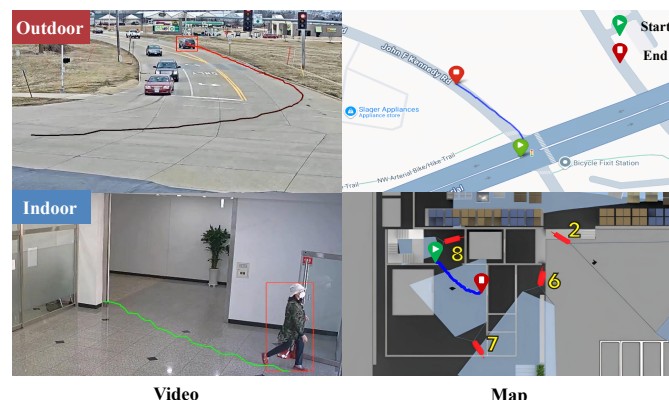

Figure 9: Camera Calibration          Figure 10: Trajectory Transformation

**Camera Calibration**. We perform geometric calibration for each camera to accurately map target trajectories from 2D image coordinates to real-world positions, as shown in Figure 9. This process involves computing a homography matrix that establishes a precise projective transformation between the camera's image plane and a top-down map of the environment. The matrix is derived by manually annotating a minimum of eight correspondence points between the image and the map. The resulting 3×3 homography matrix enables the transformation of a target's pixel coordinates into accurate real-world map coordinates, forming the basis for trajectory analysis.

**Trajectory Transformation**. We proceed to spatial-temporal feature derivation with the geometric mapping established, as shown in Figure 10. This process begins by projecting the target's bounding box coordinates onto the map using the calibrated homography matrix, followed by applying scaling factors to convert pixel measurements into real-world metrics such as meters. Subsequently, key motion parameters—including distance, velocity, and direction—are algorithmically computed for each timestamp. The resulting trajectory data undergoes a rigorous cleaning and filtering stage, where we remove anomalous points, apply interpolation for continuity, and implement multi-layer validation for temporal, spatial, and identity consistency. Finally, the quantitative motion data is synthesized into qualitative textual descriptions using a large language model, producing a "Motion Summary" that provides both numerical and narrative insights for each trajectory segment.

## A.2 ANALYSIS OF CALIBRATION DATA AND QUALITY

GTR-Bench builds upon two well-established ReID benchmarks: CityFlow Tang et al. (2019) and MTMMC Woo et al. (2024). We leverage the synchronized bounding box sequences and precise temporal metadata provided by these datasets. Specifically, since the CityFlow dataset already includes a comprehensive set of ground-plane homography matrices, we utilized these existing calibrations directly. Our primary annotation effort was directed toward generating high-quality homography matrices for the MTMMC dataset.

To validate the precision of our spatial alignment, we conducted a rigorous reprojection error analysis across all camera views. The quantitative results are summarized in Table 6.

| Scene (Dataset) | Statistic | Error (pixels) | Error (meters) |
|---|---|---|---|
| Outdoor (CityFlow) | Average | 13.25 | 2.18 |
| | Minimum | 2.19 (S04_c039) | 0.20 (S04_c031) |
| | Maximum | 21.26 (S04_c024) | 14.52 (S04_c029) |
| Indoor (MTMMC) | Average | 12.83 | 1.18 |
| | Minimum | 1.37 (c04) | 0.12 (c04) |
| | Maximum | 35.06 (c16) | 3.12 (c16) |

Table 6: Reprojection Error Analysis for CityFlow and MTMMC Datasets.

**Error Margin Analysis.** In outdoor scenarios, the mean metric error is 2.18m. Relative to the average inter-camera distance of 984.77m, the relative error is restricted to approximately 0.22%. For indoor scenes, the mean error of 1.18m represents a 4% deviation relative to the 30.59m average baseline. The spatial perturbation scale of our designed distractors exceeds 5m for indoor settings and 15m for outdoor settings. Thus, the observed reprojection errors remain well below the threshold of noise.

**Manual Verification.** While certain cameras (e.g., S04_c024 in CityFlow and c16 in MTMMC) exhibit higher localized reprojection errors due to perspective distortion, we have implemented a manual verification stage. This ensures that any potential error propagation is mitigated, maintaining the overall geometric integrity of GTR-Bench.

### A.3 MORE DETAILS ON TASK CONSTRUCTION

**Map Generation and Video Annotation**. The foundation of our task construction pipeline rests on two core components: detailed environmental maps and corresponding video data. Depending on the scenario, we utilize either indoor floor plans or outdoor road network maps, each richly annotated with essential camera deployment details, including their precise coordinates, field-of-view (FOV) coverage, viewing orientation, and references for scale and direction. Complementing this spatial information, the video data consists of video clips which we extract based on target movement. Within these clips, targets are highlighted with red bounding box annotations, providing the critical visual, temporal, and behavioral context required for our comprehensive reasoning tasks.

**Distractor Option Generation**. To ensure a rigorous evaluation and prevent models from relying on superficial heuristics, we have developed a sophisticated, scenario-specific strategy for generating plausible yet incorrect distractor options. For indoor environments, distractors are sourced from cameras located on different floors or in functionally distinct areas, creating choices that are spatially diverse yet architecturally coherent. In outdoor scenarios, we algorithmically generate synthetic camera locations along the road network. These virtual cameras are endowed with realistic attributes, including strategically positioned coordinates, plausible viewing orientations, and standard field-of-view (FOV) coverage. Furthermore, to mitigate biases arising from numerical patterns, we implement an ID randomization system that reassigns camera identifiers. This forces the model to reason based on fundamental spatial and temporal relationships rather than exploiting simple numerical sequences. These carefully designed distractor strategies are crucial for ensuring that our benchmark robustly assesses genuine geo-temporal reasoning capabilities.

## B DETAILS OF DATA DISTRIBUTION

As detailed in Table 7, our benchmark features two scenarios with vastly different spatial scales to create a diverse testbed. The outdoor network spans a city block with 31 cameras and an average inter-camera distance of 984.77 meters. In contrast, the indoor network provides building-level coverage across three floors with 16 cameras and a much smaller average distance of only 30.59 meters. This significant disparity, with the outdoor scenario being approximately 32 times larger in spatial scale, establishes a robust and diverse challenge for evaluating the spatial-temporal intelligence of VLMs across both expansive and compact environments.

| Environment | Location | Cameras | Avg. Dist. (m) | Distance Range (m) | Avg. FoV (m²) | FoV Range (m²) |
|---|---|---|---|---|---|---|
| Indoor | Level-1 | 7 | 31.38 | 5.41–60.93 | 222.81 | 11.39–449.99 |
| | Level-2 | 7 | 30.81 | 4.71–56.49 | 141.93 | 5.29–407.52 |
| | Level-3 | 2 | 9.35 | 9.35–9.35 | 53.83 | 6.44–101.21 |
| | **Overall** | **16** | **30.59** | **4.71–60.93** | **166.30** | **5.29–449.99** |
| Outdoor | CityFlow S03/04/05 | 31 | 984.77 | 0.00*–2389.42 | N/A | N/A |

Indoor measurements based on pixel coordinates (0.089 m/pixel conversion).
Outdoor measurements calculated using GPS coordinates and Haversine formula.
FoV data available only for indoor cameras with detailed polygon annotations.
Zero distance indicates cameras at identical GPS coordinates with different orientations.

Table 7: Detailed Camera Network Analysis by Environment and Location

## C  MORE DETAILS OF RESULTS

### C.1  DETAILS OF THE REID BASELINE

**Implementation and Algorithm.** To establish a traditional CV reference, we implement a ReID-based baseline using a modular pipeline. For each instance, a target query image is first extracted from the context video. We then employ YOLOv8 to generate object proposals (bounding boxes) within the candidate videos. Feature matching is conducted using the SBS algorithm (an enhanced Bag-of-Tricks (Luo et al., 2019)) via the FastReid framework (He et al., 2023).

For spatial-related tasks (e.g., GL, ATI), the model identifies the camera ID by selecting the option with the highest feature similarity within the predicted time range. For temporal tasks (e.g., NSF, TF, MTTF), timestamps exceeding class-specific similarity thresholds (0.4 for vehicles and 0.96 for persons) are aggregated to form trajectory intervals. For forecasting tasks, we utilize future video segments to align with the trajectory time.

**Fairness and Task Scope.** We implement this baseline for the GL, ATI, NSF, TF, and MTTF tasks. Tasks requiring complex semantic reasoning (i.e., MS and CR) are excluded as they are inherently infeasible for pure ReID methods. It is important to note that this comparison is inherently conservative for VLMs. Standard ReID typically requires extracting visual content from ground-truth options, whereas GTR-Bench restricts access to option-associated video content to maintain a zero-shot reasoning setting.

**Analysis of Results.** As shown in Table 4, the ReID baseline achieves an average MCQ accuracy of 45.72% and a ST-IoU of 23.05%. Despite having direct access to fine-grained visual context, the ReID baseline's performance is merely comparable to the state-of-the-art Gemini-2.5-Pro (44.90% Acc). This finding validates the significant geo-temporal reasoning capabilities of VLMs, demonstrating their ability to leverage graphical maps and video context to effectively reason about and predict unobserved events beyond simple visual matching.

### C.2  ANALYSIS OF PARTIAL SPATIAL CORRECTNESS

To further investigate the interplay between spatial localization and temporal reasoning in VLMs, we evaluate the models' performance specifically on instances where they achieve correct spatial grounding. This analysis aims to disentangle whether low ST-IoU scores stem from a complete failure in reasoning or are primarily penalized by initial spatial identification errors (e.g., incorrect camera ID).

For the NSF, TF, and MTTF tasks, we define ST-IoU (Soft) as a conditional metric. It evaluates the average ST-IoU by restricting the scope to instances where the model achieves a perfect Multiple-Choice Question (MCQ) accuracy as the model accurately predicts the spatial location. As shown in Table 8, most models exhibit a substantial performance gain under this metric. For instance, InternVL3-38B and Gemini-2.5-Pro demonstrate a significant increase in ST-IoU when spatial constraints are met, suggesting that these models possess latent temporal reasoning capabilities that are often masked by spatial misalignment.

| Model Name | Outdoor | | | | | | Indoor | | | | | |
|---|---|---|---|---|---|---|---|---|---|---|---|---|
| | NSF | | TF | | MTTF | | NSF | | TF | | MTTF | |
| | ST-IoU | SC* | ST-IoU | SC* | ST-IoU | SC* | ST-IoU | SC* | ST-IoU | SC* | ST-IoU | SC* |
| Qwen2-VL-7B-Instruct | 5.78 | 13.33 | 0.00 | 0.00 | 10.01 | 19.36 | 3.62 | 13.59 | 0.00 | 0.00 | 0.00 | 0.00 |
| Qwen2-VL-2B-Instruct | 0.00 | 0.00 | 0.28 | 0.00 | 0.00 | 0.00 | 0.00 | 0.00 | 0.21 | 3.09 | 0.00 | 0.00 |
| Qwen2.5-VL-7B-Instruct | 0.00 | 0.00 | 0.00 | 0.00 | 0.51 | 3.09 | 0.00 | 0.00 | 0.00 | 0.00 | 0.00 | 0.00 |
| Qwen2.5-VL-32B-Instruct | 0.65 | 1.63 | 0.00 | 0.00 | 15.72 | 31.43 | 3.33 | 33.33 | 0.00 | 0.00 | 0.00 | 0.00 |
| InternVL3-8B | 0.00 | 0.00 | 4.79 | 0.00 | 5.42 | 14.79 | 0.00 | 0.00 | 0.79 | 0.00 | 1.67 | 25.05 |
| InternVL3-38B | 8.27 | 35.46 | 8.20 | 52.45 | 20.58 | 41.17 | 11.10 | 47.59 | 4.37 | 0.00 | 10.24 | 61.45 |
| InternVL3-2B | 6.15 | 18.46 | 0.00 | 0.00 | 9.95 | 36.07 | 0.65 | 9.78 | 0.08 | 0.00 | 1.62 | 45.30 |
| GPT-5 | 12.04 | 14.21 | 12.12 | 0.00 | 7.34 | 15.40 | 11.34 | 27.58 | 2.55 | 0.00 | 1.75 | 15.30 |
| GPT-4o | 20.53 | 38.49 | 0.00 | 0.00 | 23.10 | 30.13 | 13.00 | 43.32 | 0.00 | 0.00 | 2.79 | 20.94 |
| GLM-4.1V-9B-Thinking | 10.29 | 27.44 | 0.00 | 2.72 | 25.38 | 33.11 | 2.87 | 27.74 | 0.00 | 0.00 | 1.67 | 25.05 |
| Gemini-2.5-Pro | 19.13 | 49.75 | 13.16 | 8.42 | 19.18 | 37.07 | 25.11 | 57.94 | 28.09 | 46.87 | 14.37 | 39.19 |
| Claude-3.5-Sonnet | 8.05 | 34.12 | 6.18 | 0.00 | 16.94 | 25.67 | 2.60 | 9.00 | 4.01 | 17.68 | 0.00 | 0.00 |
| Claude-3.7-Sonnet | 25.75 | 70.23 | 8.90 | 24.31 | 21.97 | 29.96 | 9.48 | 40.62 | 9.51 | 49.45 | 3.56 | 53.35 |

* SC denotes the ST-IoU (Spatially Correct) metric, which evaluates temporal accuracy on subsets where the model correctly identifies the spatial location.

Table 8: Comparison of ST-IoU and Spatially Correct ST-IoU.

## C.3  ANALYSIS OF SOFT ST-IoU

To more comprehensively evaluate the spatial-temporal reasoning capabilities of VLMs, particularly in cases where models predict cameras within a reasonable proximity to the ground truth, we introduce a *Soft* version of the ST-IoU metric. The Soft ST-IoU employs a stepped scoring mechanism based on the Euclidean distance, $d$, between each predicted camera $C_{p_i}$ and the ground-truth camera $C_{gt_i}$.

| Distance ($d$) | Soft Score |
|---|---|
| $d < scale$ | 1.0 |
| $scale \leq d < 2 \cdot scale$ | 0.5 |
| $2 \cdot scale \leq d < 3 \cdot scale$ | 0.2 |
| $d \geq 3 \cdot scale$ | 0.0 |

Table 9: Stepped scoring mechanism for Soft ST-IoU.

We define a base distance, termed $scale$, to account for the environmental context: 5m for indoor scenes (representing the average radius of a room) and 15m for outdoor scenes (reflecting the effective visual context in outdoor street buildings). The soft spatial score $S(C_{p_i}, C_{gt_i})$ is assigned based on this distance, as detailed in Table 9.

The formula for the Soft ST-IoU is defined as the average weighted temporal overlap across $N$ samples:

$$\text{ST-IoU}_{\text{soft}} = \frac{1}{N} \sum_{i=1}^{N} S(C_{p_i}, C_{gt_i}) \times \frac{|T_{p_i} \cap T_{gt_i}|}{|T_{p_i} \cup T_{gt_i}|} \qquad (2)$$

where $T_{p_i}$ and $T_{gt_i}$ represent the predicted and ground-truth time intervals, respectively.

Experimental results in Table 10 indicate a universal improvement in scores when using the Soft ST-IoU metric, with the Temporal Findings (TF) task exhibiting the most significant gains. This suggests that while models often identify the correct temporal window, their spatial localization frequently falls within a near-miss radius of the ground truth. Crucially, we compared the model rankings between ST-IoU and ST-IoU (soft) and found the overall hierarchy remained largely consistent, with only minor fluctuations. This stability confirms that GTR-Bench effectively benchmarks the Geo-Temporal Reasoning capabilities across different models and provides a robust evaluation across varying levels of spatial precision.

| Model | GTR-Outdoor | | | | | | GTR-Indoor | | | | | | Average | | Rank | |
|---|---|---|---|---|---|---|---|---|---|---|---|---|---|---|---|---|
| | NSF | | TF | | MTTF | | NSF | | TF | | MTTF | | | | | |
| | Std. | Soft | Std. | Soft | Std. | Soft | Std. | Soft | Std. | Soft | Std. | Soft | Std. | Soft | Std. | Soft |
| Gemini-2.5-pro | 19.13 | 21.87 | 13.16 | 15.83 | 19.18 | 19.18 | 25.11 | 29.71 | 28.09 | 30.73 | 14.37 | 20.74 | 19.84 | 23.01 | 1 | 1 |
| Claude-3-7-sonnet-latest | 25.75 | 26.37 | 8.90 | 26.79 | 21.97 | 21.97 | 9.48 | 18.53 | 9.51 | 21.54 | 3.56 | 5.84 | 13.19 | 20.17 | 2 | 2 |
| InternVL3-38B | 8.27 | 8.27 | 8.20 | 18.51 | 20.58 | 20.58 | 11.10 | 12.95 | 4.37 | 11.54 | 10.24 | 11.93 | 10.46 | 13.97 | 3 | 4 |
| GPT-4o | 20.53 | 21.76 | 0.00 | 18.79 | 23.10 | 23.10 | 13.00 | 16.65 | 0.00 | 16.40 | 2.79 | 4.58 | 9.90 | 16.88 | 4 | 3 |
| GPT-5 | 12.04 | 14.97 | 12.12 | 13.55 | 7.34 | 7.92 | 11.34 | 10.62 | 2.55 | 10.22 | 1.75 | 4.90 | 7.86 | 10.36 | 5 | 8 |
| GLM-4.1V-9B-Thinking | 10.29 | 10.83 | 0.00 | 14.83 | 25.38 | 25.38 | 2.87 | 8.45 | 0.00 | 13.44 | 1.67 | 2.00 | 6.70 | 12.49 | 6 | 5 |
| Claude-sonnet-4 | 8.05 | 11.46 | 6.18 | 15.95 | 16.94 | 18.21 | 2.60 | 6.11 | 4.01 | 16.41 | 0.00 | 2.09 | 6.30 | 11.71 | 7 | 6 |
| Qwen2.5-VL-32B-Instruct | 0.65 | 12.55 | 0.00 | 20.52 | 15.72 | 19.84 | 3.33 | 7.11 | 0.00 | 7.00 | 0.00 | 2.87 | 3.28 | 11.65 | 8 | 7 |
| Qwen2-VL-7B-Instruct | 5.78 | 9.11 | 0.00 | 6.83 | 10.01 | 10.77 | 3.62 | 9.88 | 0.00 | 2.00 | 0.00 | 1.11 | 3.24 | 6.62 | 9 | 10 |
| InternVL3-2B | 6.15 | 6.15 | 0.00 | 11.50 | 9.95 | 9.95 | 0.65 | 6.15 | 0.08 | 2.34 | 1.62 | 2.68 | 3.08 | 6.46 | 10 | 11 |
| InternVL3-8B | 0.00 | 0.00 | 4.79 | 8.10 | 5.42 | 5.42 | 0.00 | 0.00 | 0.79 | 4.52 | 1.67 | 1.67 | 2.11 | 3.29 | 11 | 13 |
| Qwen2.5-VL-7B-Instruct | 0.00 | 9.95 | 0.00 | 10.86 | 0.51 | 2.92 | 0.00 | 2.14 | 0.00 | 1.07 | 0.00 | 0.31 | 0.09 | 4.54 | 12 | 12 |
| Qwen2-VL-2B-Instruct | 0.00 | 15.70 | 0.28 | 6.48 | 0.00 | 15.71 | 0.00 | 2.35 | 0.21 | 0.21 | 0.00 | 0.34 | 0.08 | 6.80 | 13 | 9 |

Table 10: Performance comparison on GTR-Bench using standard ST-IoU (Std.) and Soft ST-IoU (Soft.) metrics.

## C.4 ABLATION EXPERIMENT OF MAP CONTEXT

We utilized graphic maps as prompts to maintain spatial consistency across cameras. To validate the effectiveness of this design in preserving inter-camera spatial relationships within our dataset construction pipeline, we conducted a ablation study on map context.

**Prompt Design of Map Context.** We elaborate on how our prompt design endows the model with rich spatio-temporal context by incorporating spatial relationships through two primary mechanisms, as Figure 11 and 12:

- **Graphic Map Context:** We utilize a rendered map image based on real-world geographical information from OpenStreetMap. This map precisely annotates the locations, orientations, and FoV of all candidate cameras. Furthermore, we explicitly included a north-arrow and a scale bar on the map to provide the model with absolute geographical orientation and scale references.

- **Textual Map Context:** As a supplement and abstraction of the map information, we designed a text-only prompt to describe the spatial relationships between cameras. This approach translates topological relations and relative positions (e.g., Camera A is located 50 meters northeast of Intersection B, facing southeast) into structured natural language descriptions.

**Construction of Textual Map Context.** We constructed the textual map context using distinct strategies for indoor and outdoor environments, tailored to their specific spatial characteristics:

- **Indoor Scenarios:** Given the smaller spatial span and finite number of cameras in indoor settings, we employed a manual annotation process structured into three hierarchical levels:
  1. *Region and Connectivity:* We divided the environment into distinct functional regions and identified key spatial connection points—such as entrances, exits, and stairwells—to define the topology.
  2. *Camera Geographic Attributes:* We described the camera's location, orientation, and FoV. The FoV corresponds to the visual representation on the map, specifically the blue area enclosed by a thin black line originating from the red camera icon.
  3. *Camera View Attributes:* We provided a detailed description of the camera's actual field of view, summarizing the visual content observed in the video from camera.

- **Outdoor Scenarios:** Due to the extensive spatial coverage of outdoor environments, we adopted an automated approach leveraging geospatial data. Using the latitude and longitude coordinates of each camera, we queried *OpenStreetMap* to retrieve the surrounding road network topology. The resulting textual context integrates information regarding roads, intersections, and specific camera metadata to reconstruct the outdoor spatial environment.

**Results and Analysis.** We benchmarked two SOTA models, Gemini-1.5-pro and InternVL-38B, on GTR-Bench. The quantitative results are presented in Table 11.

**Graphic Map Context**

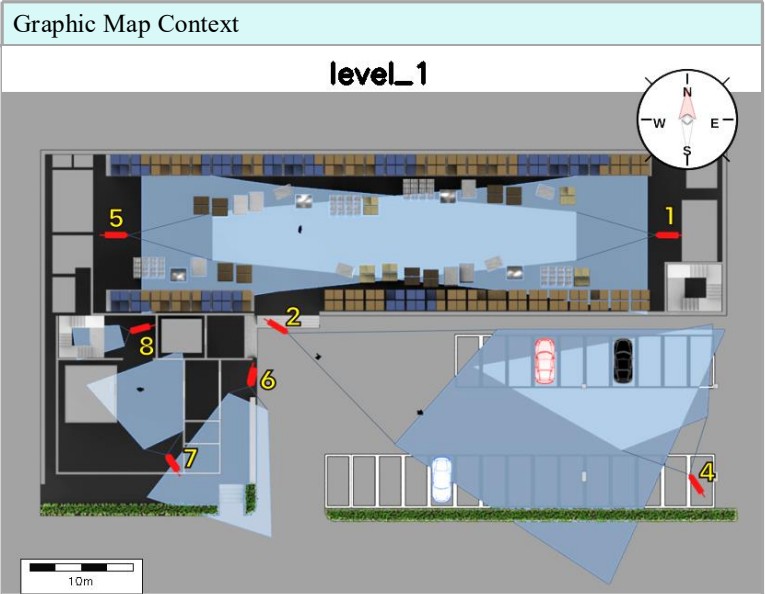

**Textual Map Context**

**Primary Areas & Key Connection**:

**The primary areas**
- a large Warehouse occupying the northern portion of the level, extending approximately 60 meters east-west and 30 meters north-south;
- a Lobby situated in the southwest quadrant, measuring roughly 15 meters by 20 meters;
- an outdoor Parking Lot to the southeast, spanning approximately 40 meters east-west and 25 meters north-south.

**Key connection points**
- a door at the south end of the Warehouse's west wall (approximately 10 meters from the southwest corner) that leads directly into the Left Stairwell; immediately east of this door, an external entrance exits to the Parking Lot;
- the south end of the Left Stairwell connects to the Lobby via a doorway positioned about 5 meters west of the Lobby's southeast corner;
- the Lobby's eastern entrance opens directly into the Parking Lot, located approximately 15 meters east of the stairwell access point;
- the southeast corner of the Warehouse connects to the Right Stairwell via a doorway, which also links to the northeast corner of the Parking Lot through an additional entrance roughly 10 meters east of the stairwell base.

**Camera Location and Abstract View**:
- Camera 1 is mounted on the east wall of the Warehouse, near its southeast corner, approximately 5 meters north of the door to the Right Stairwell. Its field of view (FOV), depicted as a blue sector, extends northwestward across the central aisle of the Warehouse, covering a span of roughly 35 meters and capturing activity along the southern storage racks.
- Camera 2 is positioned at the transition point between the Lobby's eastern exit and the Parking Lot, facing southeast. Its FOV spans approximately 25 meters diagonally across the parking area, encompassing the entrance zone and the first two rows of parking spaces.
- Camera 4 is located at the northeast corner of the Parking Lot, near the entrance connecting to the Right Stairwell, oriented southwestward. Its FOV covers the entire width of the lot's northern section, extending about 30 meters southwest to include three parked vehicles and the adjacent walkway.
- .....

**Camera Concrete FoV** :
- Camera 1 FOV: This camera (ID: 1) overlooks a wide central aisle in a large warehouse. The view spans the main corridor running north-south, flanked by tall storage racks on both east and west sides. Stacks of palletized boxes occupy the floor space along the aisle, with workstations or staging areas visible at the far end (north). No named roads or external areas are visible — it is entirely an indoor industrial storage space.
- Camera 2 FOV: Camera ID 2 covers a multi-level outdoor parking lot adjacent to a modern building with an open garage entrance marked "3". The view spans the central paved walkway between parked vehicles, extending toward a street lined with orange traffic cones and bordered by trees and a second building. No named roads are visible, but the scene includes pedestrian pathways within the parking area and access to an underground or covered garage structure. Directionally, the camera faces southeast, capturing movement from the garage exit toward the outer lot and roadside.
- Camera 4 FOV: Camera 4 covers a covered outdoor parking area adjacent to a building entrance. It captures a row of parked vehicles (white SUV, black SUV, gray sedan) aligned along marked parking spaces. The view includes a pedestrian walkway in front of the building's glass facade and extends toward an open entryway or driveway on the left, where people are entering/exiting. No named roads are visible; the scene is confined to a private or semi-private parking/entry zone of a commercial or institutional building.
- ….

Figure 11: Graphic Map vs Textual Map (Indoor)

## Graphic Map Context

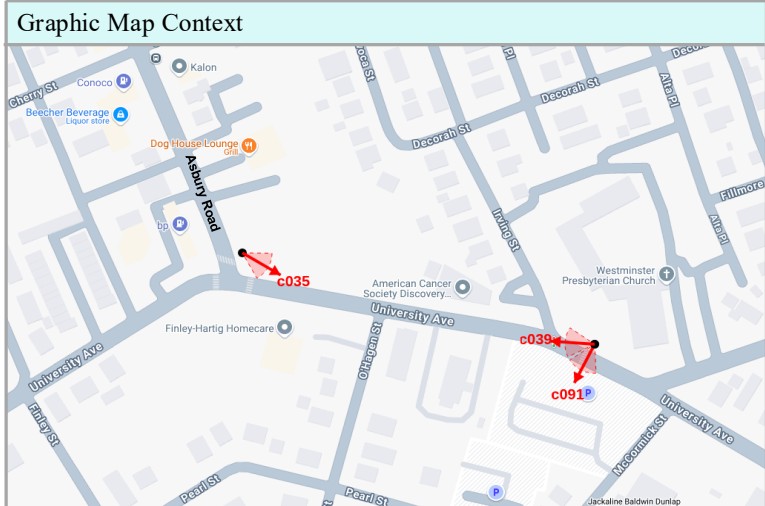

## Textual Map Context

**Map layout**:
- Asbury Road: runs Southeast-Northwest, intersects with intersects with University Avenue.
- Ashton Place: runs Southeast-Northwest, intersects with intersects with Decorah Street.
- Decorah Street: runs West-East, intersects with intersects with Ashton Place, Irving Street.
- Irving Street: runs Southeast-Northwest, intersects with intersects with Decorah Street, University Avenue.
- O'Hagen Street: runs North-South, intersects with intersects with University Avenue.
- University Avenue: runs West-East, intersects with intersects with Asbury Road, Irving Street, O'Hagen Street.

**Intersection relationships**:
- Asbury Road & University Avenue:
  - 118 meters north from intersection of University Avenue & O'Hagen Street
  - 257 meters north from intersection of University Avenue & Irving Street
  - 260 meters northeast from intersection of Ashton Place & Decorah Street
  - 199 meters northeast from intersection of Decorah Street & Irving Street
- University Avenue & O'Hagen Street:
  - 118 meters south from intersection of Asbury Road & University Avenue
  - 138 meters north from intersection of University Avenue & Irving Street
  - 170 meters northeast from intersection of Ashton Place & Decorah Street
  - 120 meters northeast from intersection of Decorah Street & Irving Street
- University Avenue & Irving Street:
  - 257 meters south from intersection of Asbury Road & University Avenue
  - 138 meters south from intersection of University Avenue & O'Hagen Street
  - 142 meters east from intersection of Ashton Place & Decorah Street
  - 146 meters southeast from intersection of Decorah Street & Irving Street
- Ashton Place & Decorah Street:
  - 260 meters southwest from intersection of Asbury Road & University Avenue
  - 170 meters southwest from intersection of University Avenue & O'Hagen Street
  - 142 meters west from intersection of University Avenue & Irving Street
  - 62 meters south from intersection of Decorah Street & Irving Street
- Decorah Street & Irving Street:
  - 199 meters southwest from intersection of Asbury Road & University Avenue
  - 120 meters southwest from intersection of University Avenue & O'Hagen Street
  - 146 meters northwest from intersection of University Avenue & Irving Street
  - 62 meters north from intersection of Ashton Place & Decorah Street

**Camera details**:
- c035:
  - Location: 17 meters northeast from intersection of Asbury Road & University Avenue
  - Orientation: facing southeast
  - Field of View: Covering Main Street westward range approximately 150 meters, capturing traffic flow and adjacent commercial properties including Finley-Hartig Homecare. The view extends from the grassy median barrier to the far side of the road, encompassing multiple lanes and parked vehicles.
- c039:
  - Location: 26 meters east from intersection of University Avenue & Irving Street
  - Orientation: facing west
  - Field of View:
- c091:
  - Location: 26 meters east from intersection of University Avenue & Irving Street
  - Orientation: facing southwest
  - Field of View: covering Main Street westward range approximately 50 meters, with visibility extending to adjacent parking lots and cross-street traffic signals.

Figure 12: Graphic Map vs Textual Map (Outdoor)

| Model | Map Type | GTR-Outdoor | | | | | | | GTR-Indoor | | | | | | |
|---|---|---|---|---|---|---|---|---|---|---|---|---|---|---|---|
| | | ATI | MS | GL | CR | NCF | TF | MTTF | ATI | MS | GL | CR | NCF | TF | MTTF |
| Gemini-Pro | image | 60.00 | 46.67 | 33.33 | 56.67 | 19.13 | 13.16 | 19.18 | 63.33 | 13.33 | 26.67 | 70.00 | 25.11 | 28.09 | 14.37 |
| Gemini-Pro | text | 43.33 | 56.67 | 60.00 | 66.67 | 6.83 | 0.00 | 1.11 | 13.33 | 43.33 | 73.33 | 83.33 | 27.76 | 19.76 | 21.42 |
| InternVL-38B | image | 40.00 | 73.33 | 30.00 | 53.33 | 8.27 | 8.20 | 20.58 | 50.00 | 56.67 | 26.67 | 37.93 | 11.10 | 4.37 | 10.24 |
| InternVL-38B | text | 38.33 | 33.33 | 76.67 | 36.36 | 3.06 | 1.08 | 3.99 | 50.00 | 33.33 | 56.67 | 73.33 | 4.36 | 0.55 | 9.20 |

Table 11: **Ablation Study of Map Context.** The table compares the performance of models using Image-based vs. Text-based map contexts across Indoor and Outdoor subsets.

The results clearly indicate that there is no single best method for encoding spatial relationships. The choice between visual (image) and textual context significantly impacts performance, and the optimal choice varies by task and environment: (1) Text-based prompts tend to excel in tasks requiring abstract or logical spatial reasoning. For instance, both models perform significantly better on the GL and CR tasks with textual context, particularly in the indoor setting. This suggests that for tasks involving topological understanding and logical inference, a structured textual description is more effectively utilized by the models. (2) Image-based prompts demonstrate a strong advantage in tasks that rely on understanding physical motion and trajectories. In GTR-Outdoor, both models show improved performance on motion-related tasks such as MS , TF , and MTTF when provided with a graphic map.

## D  PROMPT FOR CONTEXT UTILIZATION ANALYSIS

---

### Context Utilization Analysis Prompt

[Reasoning Text]
The video from camera c07 shows the target (ID 4) entering an indoor lobby area from the right side of the frame and walking towards the left. The last frame at 12:00:28.040 shows the target moving towards a hallway… the location as the southern entrance of the building…

[Rule]
Please analyze [Reasoning Text] and determine whether it contains the specified key elements. For each element, mark it as 1 if it is explicitly mentioned or referenced in the text, otherwise mark it as 0.
Camera { i }
1.  **Location**: Whether the specific position, direction, or area of camera {i} is mentioned
2.  **Time**: Whether the specific time information of camera {i} recording is mentioned
3.  **Motion**: Whether the movement, direction, or behavior of targets in camera {i} is described
Camera { i+1 }…

[Output]
Please return the result in JSON format: {"camera{i}_analysis":
{"Location": 0 or 1, "Time": 0 or 1, "Motion": 0 or 1}…}

---

Figure 13: Context Utilization Analysis Prompt

We employ the Qwen-max closed-source model to analyse the reasoning content of various models across seven tasks in GTR-Bench, examining whether the reasoning process contains the three key

elements of Location, Time, and Motion State for the cameras specified in the given questions. The objective is to evaluate the utilization rate of models for spatial, temporal and motion modal context. Notably, for Timeline Reordering tasks, since no Camera Time information is provided, Time elements are not included in the statistical analysis. The specific prompt used is shown in Figure 13.

# E  DETAILED FAILURE CASE STUDY

## E.1  DETAILED ERROR DEFINITIONS

Our detailed error analysis reveals systematic failure patterns across different task categories and model types. Based on our experimental results, we categorize the error patterns into four main dimensions: spatial reasoning errors, temporal reasoning errors, motion reasoning errors, and reasoning mechanism errors.

**Spatial Reasoning Errors**. We identify two primary categories of spatial reasoning failures:

- **Topology Error**: Models frequently fail to understand complex spatial constraints and relationships. This includes (1) constraint understanding errors, such as failing to account for road network constraints (one-way streets, restricted areas) and indoor architectural constraints (walls, obstacles); (2) connection relationship errors, where models misunderstand the connectivity between roads or indoor floor transitions.
- **FoV Alignment Error**: Models struggle to understand the correspondence between camera field-of-view (FoV) regions and their mapping to environmental maps, particularly in scenarios involving multiple cameras with overlapping coverage areas.

**Temporal Reasoning Errors**. Our analysis reveals one critical temporal reasoning limitation:

- **Time Interval Error**: Models misestimate distance, speed, or related quantities, leading to erroneous time interval estimates. Models demonstrate significant errors in temporal interval estimation, often making predictions that deviate substantially from actual time spans, particularly in scenarios involving variable movement speeds or complex temporal patterns.

**Motion Reasoning Errors**. We observe one category of motion-related reasoning failure:

- **Motion State Error**: Models fail to correctly transform motion states from video coordinates to world coordinates, leading to errors in directional and other world motion states. This coordinate transformation failure affects the prediction of future target motion trajectories.

**Reasoning Mechanism Errors**. Our analysis identifies three fundamental issues in the reasoning process itself:

- **Information Bias**: Models exhibit systematic preferences and over-reliance on certain types of information (temporal, spatial, or motion) while neglecting others. This bias leads to imbalanced reasoning where models may prioritize spatial information while ignoring temporal constraints, or vice versa.
- **Step Jump Error**: Models frequently skip crucial intermediate reasoning steps, jumping directly from observations to conclusions without proper logical progression. This results in incomplete reasoning chains that lack the necessary intermediate analysis.
- **Reasoning Inconsistency Error**: Within the same reasoning task, models often generate contradictory reasoning steps, indicating a lack of coherent internal reasoning processes and logical consistency.

## E.2  DETAILED ANALYSIS

**Benchmark-Specific Error Patterns**. Our analysis identifies several error categories that are uniquely characteristic of geo-temporal reasoning scenarios and rarely appear in conventional

spatial-temporal intelligence evaluations. These errors stem from our benchmark's distinctive features: (1) multi-camera networks spanning large geographic areas with 2-3 distinct viewpoint types (camera views and map perspective), (2) explicit temporal reasoning requirements beyond simple sequence ordering, and (3) the need to infer unobserved states between fragmented observations.

The most prevalent errors reflect these unique challenges. **Topology Error** (28.6%) and **FoV Alignment Error** (16.8%) dominate our error distribution, representing failures in understanding complex spatial constraints that are absent in single-scene benchmarks. These include road network topology (one-way streets, restricted areas), architectural constraints (walls, obstacles), and camera field-of-view mapping—challenges that only emerge when reasoning across multiple, discontinuous viewpoints. **Motion State Error** (20.4%) represents another benchmark-specific failure mode, where models struggle with coordinate system transformations and precise velocity calculations based on map scale, rather than the simple directional motion understanding required in traditional benchmarks. **Time Interval Error** (12.4%) occurs when models fail to understand how temporal constraints (too long or too short time windows) invalidate certain path predictions—a challenge that doesn't exist in implicit temporal reasoning scenarios.

| Category | Topology | FoV | Time_Int | Motion | Info_Bias | Step_Jump | Reasoning_Inc |
|---|---|---|---|---|---|---|---|
| **Overall** | 97 | 57 | 42 | 69 | 23 | 40 | 11 |
| **Percentage** | 28.6% | 16.8% | 12.4% | 20.4% | 6.8% | 11.8% | 3.2% |
| **By Task Type:** | | | | | | | |
| Geo-location | 22 | 6 | 0 | 6 | 12 | 10 | 5 |
| Arrival Time-Interval | 7 | 17 | 12 | 1 | 3 | 4 | 0 |
| Motion-State | 10 | 11 | 11 | 13 | 5 | 4 | 3 |
| Causal Reordering | 9 | 9 | 0 | 17 | 2 | 7 | 0 |
| Next Spot Forecasting | 16 | 4 | 2 | 12 | 1 | 4 | 2 |
| Trajectory Forecasting | 20 | 6 | 9 | 4 | 0 | 8 | 1 |
| Multi-Target Trajectory Forecasting | 13 | 4 | 8 | 16 | 0 | 3 | 0 |
| **By Scenario:** | | | | | | | |
| Outdoor | 50 | 30 | 20 | 37 | 10 | 23 | 6 |
| Indoor | 46 | 27 | 22 | 32 | 13 | 16 | 5 |
| **By Model:** | | | | | | | |
| Gemini-2.5-Pro | 26 | 25 | 20 | 30 | 3 | 1 | 4 |
| GPT-4o | 34 | 19 | 14 | 22 | 5 | 4 | 4 |
| InternVL3-38B | 37 | 13 | 8 | 17 | 15 | 35 | 3 |

Table 12: Detailed error statistics. Statistics of 177 cases, each error type is counted independently, resulting in a total error count of 339 across all categories.

### E.3 DETAILED FoV ALIGNMENT ERROR

To provide a more in-depth analysis of Map-Video Alignment Errors, specifically **FoV Alignment Error**, we first distinguish between two forms of Field of View (FoV) defined in GTR-Bench.

**FoV to Map Misalignment.** This error type represents a conventional failure where the model identifies an object in the video but fails to locate its corresponding position on the map. As illustrated in Figure 14, the model successfully reasons from the video content that the target enters a "stairwell" (observed in camera c07). However, it fails to align this visual feature with the map structure, specifically missing that this stairwell corresponds to the location of the ground-truth camera c08. Consequently, the model provides an incorrect answer despite correct visual perception.

**FoV Misinterpretation.** This error occurs when the model struggles to interpret the geometric representations of FoVs on the map, particularly when multiple FoVs overlap. Figure 15 depicts a scenario where the model fails to understand the spatial relationship between cameras. For example, when analyzing a trajectory involving overlapping cameras (e.g., moving from c12 through c09 to c14), the model may incorrectly assume disjoint time intervals. This stems from a failure to recognize that the FoV of the intermediate camera (c09) spatially overlaps with the destination camera (c14), leading to logical errors in trajectory or timeline prediction.

In summary, while current VLMs encounter challenges with both alignment types, the misinterpretation of FoVs is a more prevalent and critical issue in geographic reasoning tasks involving complex map topologies.

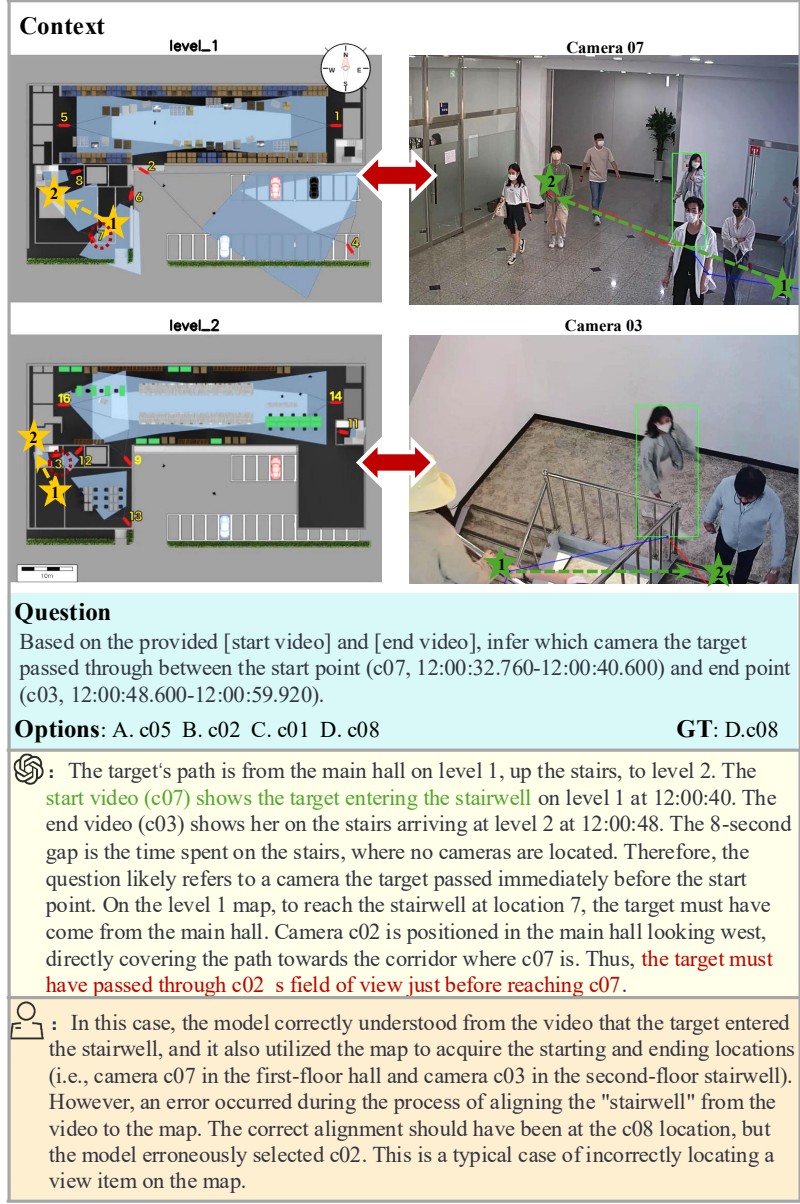

Figure 14: FoV to Map Misalignment. The model correctly identifies the semantic location ("stairwell") from the concrete video view but fails to map it to the geometric location on the map.

# F  DETAILED HUMAN ANNOTATION AND EVALUATION PROTOCOL

To ensure the reliability of our human evaluation results, we implemented a rigorous screening and training process for the personnel involved in data annotation and performance assessment.

## F.1  COMPOSITION OF QUESTION ANNOTATORS AND BENCHMARK EVALUATORS

We established strict qualification criteria and organizational protocols to maintain high data quality and evaluation independence:

- **Professional Field**: The dataset was constructed by researchers with profound academic backgrounds in 3D vision.

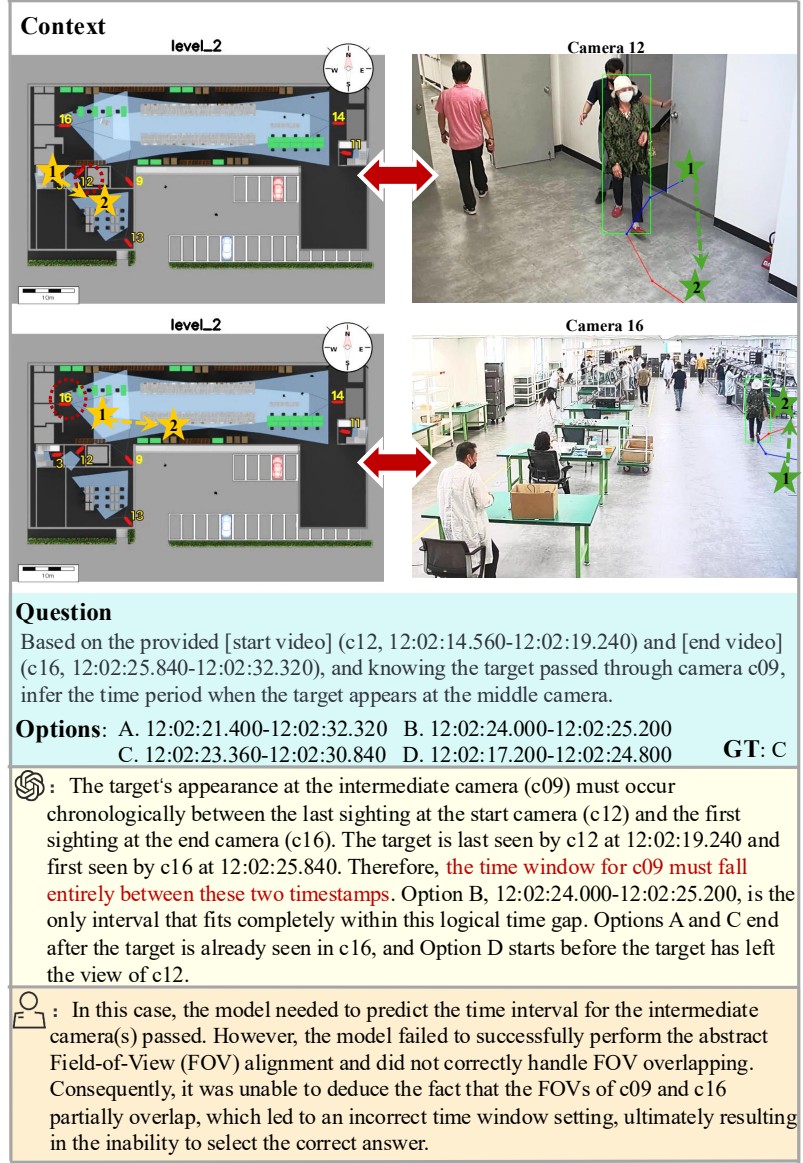

Figure 15: FoV Misinterpretation. The model fails to perceive the overlap between the Abstract FoVs of adjacent cameras on the map, leading to incorrect temporal or spatial reasoning.

- **Professional Experience**: These researchers possess extensive experience in multi-view geometry, scene understanding, and spatial intelligence.

- **Knowledge Requirements**: All were required to have a deep understanding of core concepts such as spatial relationships, camera poses, and object motion, and be capable of complex spatial reasoning.

- **Distinction between Annotators and Evaluators**: The personnel responsible for question annotation were a separate group from those who conducted the benchmark evaluation, ensuring the independence of the evaluation results.

### F.2 ANNOTATION GUIDELINES

To maintain high standards of data integrity and annotation quality, we implemented the following guidelines:

- **Multi-reviewer Mechanism**: During the dataset construction, each sample generated by our rules was cross-validated by multiple other independent annotators.

- **Dependence on Maps and Camera Videos**: We ensured that questions could not be answered solely through the map, camera video, or common sense, but required the synthesis of information from all provided map and video inputs.

- **Clarity and Unambiguity of Context and Questions**: We ensured that the content of the map images and camera videos was clear and that the question text was precise and unambiguous.

- **Uniqueness of Answer**: We ensured that there was only one correct answer and that the distractors were designed to be plausible.

- **Spatio-Temporal Span Coverage**: All seven question types in the benchmark were required to have diversity in their temporal and spatial spans to ensure the variety of the problems across different spatio-temporal dimensions.

- **Ethics and Licensing**: We ensured that all videos and maps were either fully copyrighted or used with the author's consent, and we maintained the same strict standards for personal privacy information as the reference datasets.

### F.3 EVALUATION GUIDELINES

The final human performance score was determined by multiple independent evaluators, and their average accuracy was taken. This multi-evaluator mechanism effectively reduces the randomness caused by individual differences, making the evaluation results more statistically significant and stable.

### F.4 CONSISTENCY WITH MODEL CONSTRAINTS

During the evaluation, evaluators had access only to the exact same information as the model input, namely the map, the camera videos and the question text. They were strictly forbidden from using any external knowledge or tools for assistance.

To maximize the evaluators' ability to utilize visual information, we also designed a flexible and professional front-end evaluation page, as shown in Figure 16. This interface was built to: (1) Display Map and Camera Information: Fully present all map and camera information necessary to answer the question.(2) Enable Interaction: Allow evaluators to zoom in on and draw on the images from both contexts to mark key elements during their reasoning process.This design ensured that evaluators could fully leverage the provided visual information for accurate reasoning and decision-making.

We provided human evaluators with unrestricted evaluation time, which differs from the constraints set for the model evaluation. This was primarily because our goal was to measure the optimal performance of humans under ideal conditions, thereby more clearly revealing the gap between the current model's capabilities and human abilities in complex spatial reasoning.

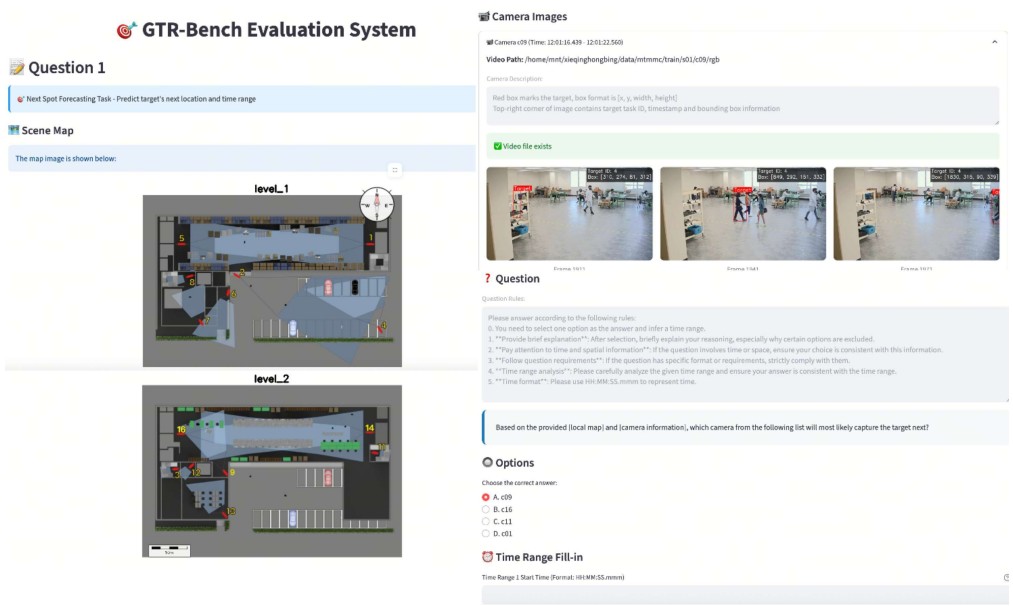

Figure 16: Evaluation System. The interface is designed to facilitate human reasoning by fully presenting necessary map and camera information. It enables interaction, allowing evaluators to zoom in and draw on the images to mark key elements during the reasoning process.

## G  MORE EXAMPLES

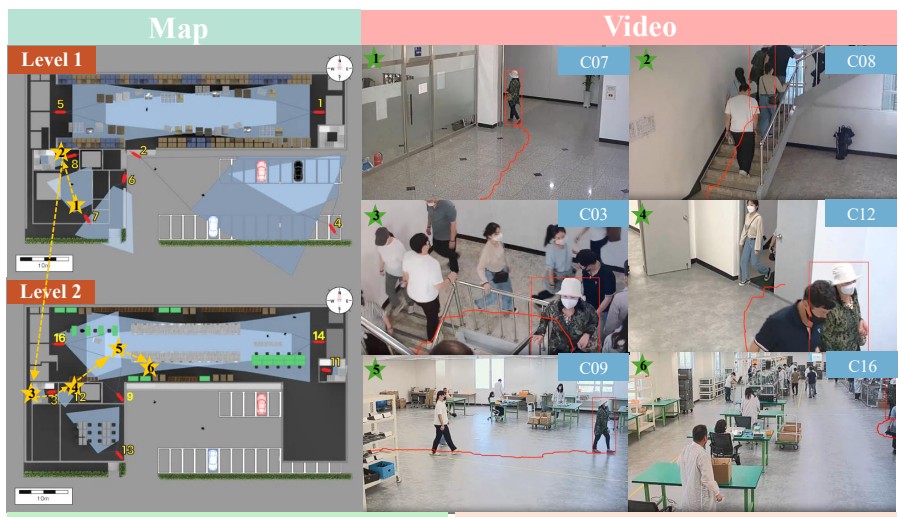

**Geo-Location**

**Question**: Based on the provided [start video⭐] and [end video⭐], infer which camera the target passed through between the start point (c07, 12:01:28.160-12:01:36.280) and end point (c03, 12:01:50.560-12:02:04.800).
**Options**: A. c13    B. c04    C. c08    D. c11

**Arrival Time-Interval**

**Question**: Based on the provided [start video⭐] (c07, 12:01:28.160-12:01:36.280) and [end video⭐] (c03, 12:01:50.560-12:02:04.800),and knowing the target passed through c08, infer when the target arrived at the intermediate camera.
**Options**: A. 12:01:35.160-12:01:47.800
               B. 12:01:34.110-12:01:40.560
               C. 12:01:47.900-12:01:48.740
               D. 12:01:34.900-12:01:49.200

**Motion-State**

**Question**: Based on the provided [start video⭐] (c12, 12:02:14.560-12:02:19.240) and [end video.⭐] (c16, 12:02:25.840-12:02:32.320),and the intermediate camera c09, infer the target's motion state during the intermediate time period.
**Options**:
A. The target moved south for 2.1 meters…

B. The target moves northeast at a speed of 1.6 m/s for 4.4 meters, then moves east at a speed of 1.5 m/s for 1.8 meters, and finally moves northeast again at a speed of 1.0 m/s for 1.6 meters.

C. The target quickly moves 6.1 meters to the northeast at a speed of 3.9 m/s…

D. …a distance of 1.9 meters…a distance of 0.6 meters.

**Causal Reordering**

**Question**: Based on the provided [local map] and [videos⭐ ⭐ ⭐], analyze the target's activity trajectory. Please infer the correct order in which the target passed through these cameras.
**Options**:
    A. c16 → c09 → c12      B. c09 → c12 → c16
    C. c12 → c01 → c16      D. c12 → c09 → c16

**Next Spot Forecasting**

**Question**: Based on the provided [local map] and [video⭐], which camera from the following list will most likely capture the target next? You need to select one option as the answer and infer a time range.
**Options**: A. c05    B. c02      C. c06      D. c08
**GT**: D. c08 12:01:35.160-12:01:47.800

**Trajectory Forecasting**

**Question**: Based on the provided [local map] and [videos ⭐ ⭐ ], predict the next two cameras that the target will likely pass through. You need to select a correct option sequence and infer a time range sequence simultaneously.
**Options**:: A. c09      B. c14      C. c13      D. c03      E. c12
**GT**: E. c12 12:02:14.560-12:02:19.240
         A. c09 12:02:23.360-12:02:30.840

**Multi-Target Trajectory Forecasting**

**Question**: Based on the provided [local map] and [videos.⭐ ⭐] showing the movement trajectories of two [Target], predict where and when these [Target] person will most likely meet.
**Options**:
    A. c14      B. c03
    C. c09      D. c12
**GT**: B. c03
12:02:03.480-12:02:04.800

Figure 17: Indoor Examples

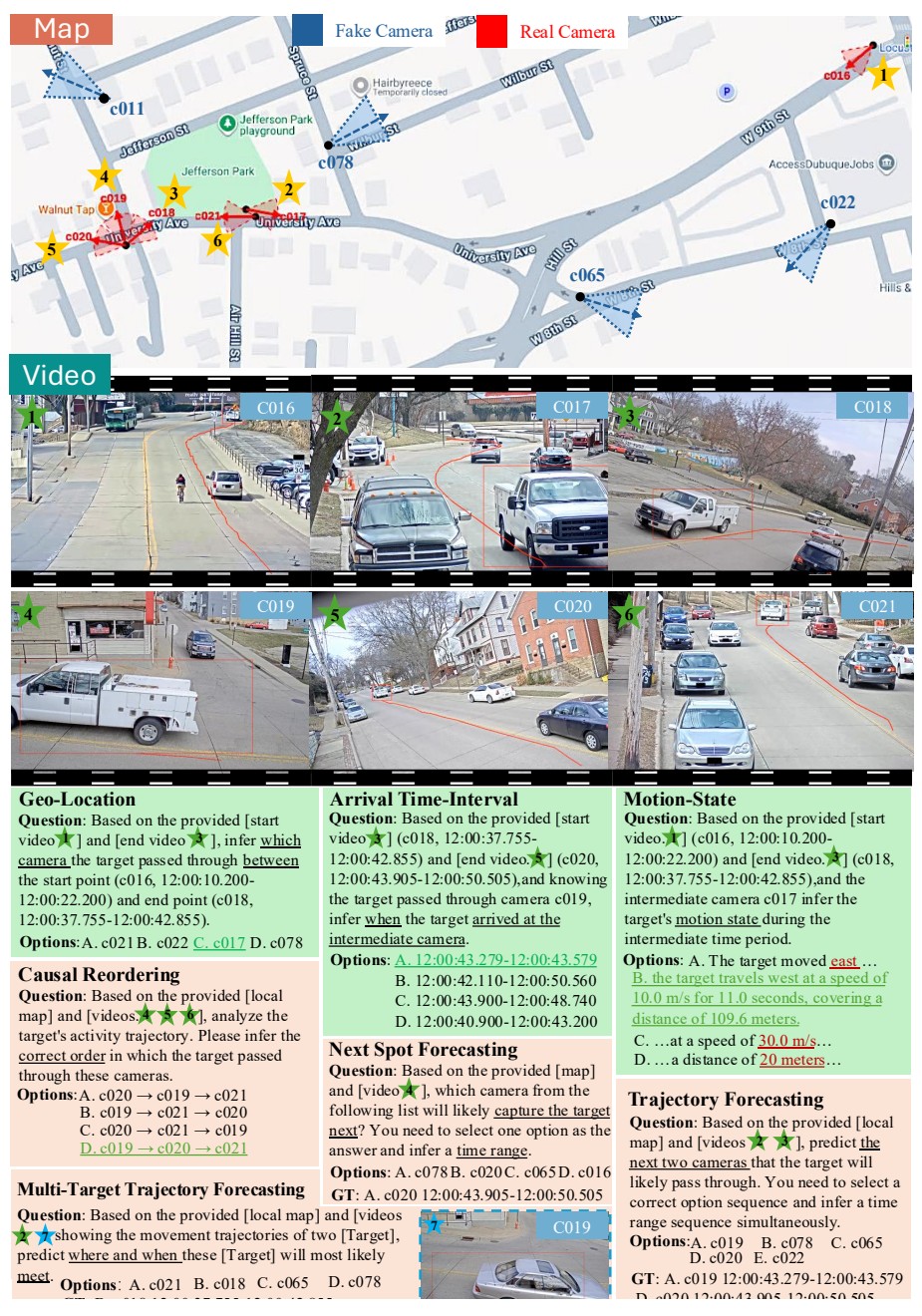

Figure 18: Outdoor Examples

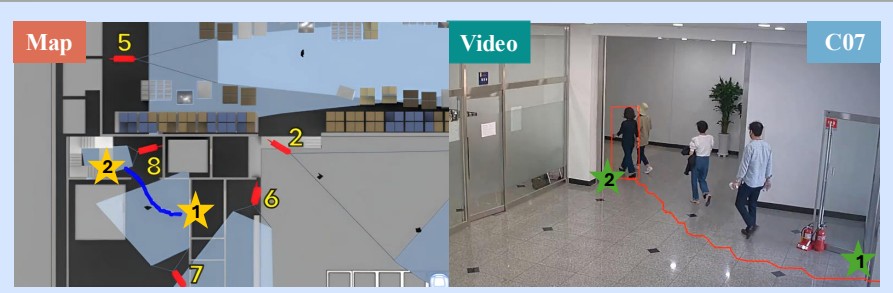

**Question**: Based on the provided [local map] and [camera information], which camera from the following list will most likely capture the target next? You need to select one option as the answer and infer a time range.

**Options**: A. c05  B. c02  C. c06  D. c08        **GT**: D. c08 12:00:26.600-12:00:37.040

🧠 **Reasoning**:

1. **Analyze Current Location and Trajectory**: The video from camera c07 shows the target (ID 4) entering an indoor lobby area from the right side of the frame and walking towards the left. The last frame at 12:00:28.040 shows the target moving towards a hallway. By comparing the visual elements in the video (glass doors, open hallway) with the map, we can identify the location as the southern entrance of the building. The target_s path is from this entrance (covered by c07) moving north-west into the building.

2. **Evaluate Potential Next Cameras**:

    **D. c08**: The map shows camera c08 is located in the hallway directly adjacent to the lobby where c07 is. Its field of view covers the path the target is taking. This makes it the immediate and most probable next camera.

    **A. c05**: This camera is located in the large hall to the north. To be seen by c05, the target must first pass through the area covered by c08.

    **B. c02**: This camera is outside in the parking lot. The target is inside and moving further into the building, away from c02.

    **C. c06**: This camera is also outside, near the entrance. The target would have likely been in its view before entering the building and being captured by c07, not after.

3. **Time Estimation**: The last known timestamp for the target in c07_s view is 12:00:28.040. Since the field of view of c08 is contiguous with c07_s along the target_s path, the target will enter c08_s view almost immediately. A reasonable estimated time range for the capture by c08 would start right after the last sighting, approximately from 12:00:28 to 12:00:38, allowing a few seconds to traverse the camera_s view.

**Answer**: D. c08  12:00:28-12:00:38                          **ST-IoU**: 0.87

Figure 19: Correct Example

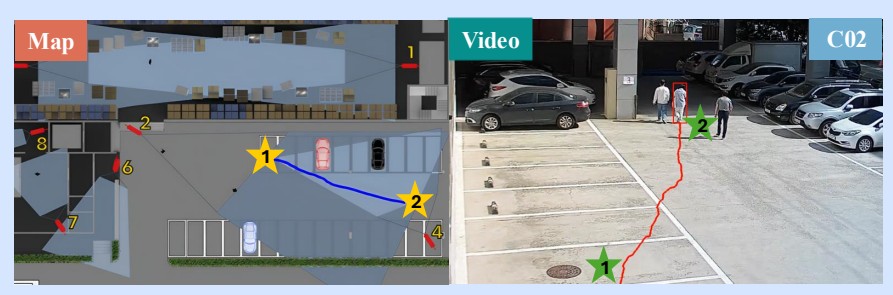

**Question**: Based on the provided [local map] and [camera information], which camera from the following list will most likely capture the target next? You need to select one option as the answer and infer a time range.
**Options**: A. c04  B. c06  C. c01  D. c02          **GT**: A.c04  12:03:09.680-12:03:18.400

🔷 **Reasoning**:
1. **Initial Observation and Target Movement**
The provided video frames are from camera c02, which is positioned on the building and overlooks the outdoor parking lot. The target (ID: 10) is observed walking from the open area of the parking lot towards the building entrance on the west side. The timestamps from 12:02:57 to 12:03:14 show the target getting progressively further from the camera and moving towards the left of the frame, which corresponds to a north-westerly direction on the map.

2. **Current Position Analysis and Next Camera Prediction**
The target's last known position at 12:03:14 is at the edge of c02's field of view, right before the building entrance. According to the map, camera c06 is positioned to monitor this specific entrance area. Therefore, as the target continues on their path, they will immediately enter the field of view of camera c06. Camera c04 is on the far eastern side of the parking lot, and the target is walking away from it. Camera c01 is inside the main building at the eastern end, which is not along the target's path. Thus, c06 is the only logical next camera.

3. **Time Estimation**
Given the last timestamp of 12:03:14, the target will enter c06's view within a few seconds. A time range of 12:03:15 to 12:03:30 is a reasonable estimate for the duration of the capture.

**Answer**: B. c06  12:03:15-12:03:30                    **ST-IoU**: 0

Figure 20: Incorrect Example

