# OpenReview forum: "GTR-Bench: Evaluating Geo-Temporal Reasoning in Vision-Language Models"
_ICLR.cc/2026/Conference — ICLR 2026 Poster_

### Official Review · Reviewer_f47F · 2025-10-26

**Soundness:** 3
**Presentation:** 3
**Contribution:** 3
**Rating:** 4
**Confidence:** 4

**Summary:**

GTR-Bench introduces a geo-temporal reasoning benchmark that tests VLMs on jointly understanding maps and multi-camera videos across indoor/outdoor scenes, spanning seven tasks from basic (Geo-location, Arrival Time-Interval, Motion-State) to combinatorial (Causal Reordering, Next-Spot, Trajectory, Multi-Target Trajectory).

Across 12 models, the best proprietary system reaches only 34.9% vs. 78.61% human accuracy, revealing three weaknesses: imbalanced use of spatial/temporal context, weak temporal forecasting, and poor map–video alignment.

**Strengths:**

* The paper proposes a comprehensive, well-balanced benchmark. Seven geo-temporal tasks, 420 questions (60 per task) across two real-world scenarios with maps plus multi-camera videos.
* The paper proposes a standardized evaluation protocol (≤20 sampled frames, fixed decoding settings) across 12 diverse VLMs with the Spatial-Temporal IoU.

**Weaknesses:**

* The primary concern is coverage across scenarios. The benchmark has 420 questions total (60 per task; 30 indoor + 30 outdoor), across only two environments—useful but relatively small for drawing strong statistical conclusions or long-tail cases.
* Videos are down-sampled to keep the total frames across clips 20 per question, which can disadvantage models designed for long-context video reasoning and alter temporal cues.

**Questions:**

The primary concern here is the adequacy of scenario coverage. Additional clarification or examples may be needed.

---

> ### Author Response · Authors · 2025-11-23
> **Response to Reviewer f47F(1/2)**
>
> Dear Reviewer f47F,
>
> We would like to thank you for providing valuable comments, and we answer the concerns raised below.
>
> > **W1**: The primary concern is coverage across scenarios. The benchmark has 420 questions total (60 per task; 30 indoor + 30 outdoor), across only two environments—useful but relatively small for drawing strong statistical conclusions or long-tail cases.
>
> > **Q1**: The primary concern here is the adequacy of scenario coverage. Additional clarification or examples may be needed.
>
> **1.Rationale for Environment Selection.**
>
> We argue that we weigh the diversity in the spatial-temporal complexity, the motion patterns and categories of the objects more than that in background scenes in this work, since the geo-temporal reasoning capabilities of VLMs relate more closely to the spatial-temporal cues. Adjustable camera configurations would largely enrich the scenarios, and we would be happy to take some investigation within simulated enviroment in furture work. As below, we detailly demonstrate the diversity in the spatial-temporal complexity, the motion patterns and categories of the subjects in our dataset.
>
> **2.Question diversity.**
>
> We developed GTR-Bench to encompass tasks across varying levels of spatial-temporal complexity. We categorized these tasks based on trajectory length ($track_d$) and duration ($track_t$); a task is classified into a higher complexity tier if it exceeds the defined threshold for either metric. The classification criteria are as follows:
>
> | Category | Outdoor Criteria | Indoor Criteria |
> | :--- | :--- | :--- |
> | **Long Geo-temporal** | $track_d > 250m$ or $track_t > 30s$ | $track_d > 35m$ or $track_t > 120s$ |
> | **Medium Geo-temporal**| $150m < track_d \leq 250m$ or $15s < track_t \leq 30s$ | $25m < track_d \leq 35m$ or $60s < track_t \leq 120s$ |
> | **Short Geo-temporal** | $track_d \leq 150m$ and $track_t \leq 15s$ | $track_d \leq 25m$ and $track_t \leq 60s$ |
>
> For spatial complexity, 250m represents the scale of traversing multiple city blocks or distinct topological changes in outdoor scenarios; 35m denotes indoor multi-room or cross-floor trajectory. For temporal complexity: 30s and 120s correspond to the traversal time for vehicles and pedestrians (respectively) at average speeds, or serve as indicators for highly complex kinematic patterns.
>
> The distribution of tasks across these categories is summarized below:
>
> | Category | Outdoor | Indoor | Total |
> | :--- | :--- | :--- | :--- |
> | **Long Geo-temporal** | 74 | 62 | 136 |
> | **Medium Geo-temporal**| 82 | 82 | 164 |
> | **Short Geo-temporal** | 54 | 66 | 120 |
>
> Beyond spatial-temporal complexity, the benchmark ensures diversity in object and motion patterns:
> *   **Object Categories:**
>     *   **Outdoor:** Includes a variety of vehicle types (sedans, trucks, SUVs) with diverse colors.
>     *   **Indoor:** Features pedestrians of different genders, ages, and heights, with varied occupations and clothing styles.
> *   **Motion Patterns:**
>     *   **Outdoor:** Covers scenarios such as straight-line driving, intersections (crossroads, T-junctions), varying traffic flow densities, and changes in traffic light timings.
>     *   **Indoor:** Includes complex patterns like moving from indoors to outdoors, ascending/descending stairs, navigating multi-room environments, and interactions with other people or objects in the scene.
>
> We fully agree with the reviewer for broader scenario coverage. We are committed to extending this benchmark in our future research.

---

> ### Author Response · Authors · 2025-11-23
> **Response to Reviewer f47F(2/2)**
>
> > **W2**: The paper proposes a standardized evaluation protocol (≤20 sampled frames, fixed decoding settings) across 12 diverse VLMs with the Spatial-Temporal IoU.
>
> > **Q2**: Videos are down-sampled to keep the total frames across clips 20 per question, which can disadvantage models designed for long-context video reasoning and alter temporal cues.
>
> **1.Why we limit to 20 frames.**
>
> Our experimental design restricts the total video input to a cumulative 20 frames. This constraint is necessary to accommodate the high-resolution imagery required for motion state recognition within the context window limits of specific models. This design ensures that our results primarily reflect the model's ability to reason about geo-temporal relationships, rather than its proficiency in decoding long-form temporal dynamics.
>
> **2.Frame-sensitivity analysis on GTR-Bench.**
>
> To evaluate sensitivity to frame sampling budgets, we conducted a new series of experiments for Gemini-2.5-pro with three distinct settings:
> + **20-frame limit**: The baseline setting used in our original paper.
> + **32-frame limit**: We increased the sampling budget to 32 frames.
> + **Trajectory frame + motion description**: To decouple the reasoning component from temporal perception, we provided a single composite image displaying the target's full trajectory, augmented with a textual description derived from ground-truth motion data.
>
> The results are presented in the table below.
>
>   |                                       | outdoor |       |       |       |       |       |       | indoor |       |       |       |       |       |       | avg    |
> |---------------------------------------|---------|-------|-------|-------|-------|-------|-------|--------|-------|-------|-------|-------|-------|-------|--------|
> | video type                            | ATI     | MS    | GL    | CR    | NCF   | TF    | MTTF  | ATI    | MS    | GL    | CR    | NCF   | TF    | MTTF  | avg    |
> | 20-frame limit                        | 60.00   | 46.67 | 33.33 | 56.67 | 19.13 | 13.16 | 19.18 | 63.33  | 13.33 | 26.67 | 70.00 | 25.11 | 28.09 | 14.37 | 34.93  |
> | 32-frame limit                        | 36.67   | 50.00 | 46.67 | 26.67 | 16.25 | 6.11  | 4.72  | 16.67  | 30.00 | 56.67 | 70.00 | 19.92 | 11.35 | 12.41 | 28.87  |
> | trajectory frame + motion description | 43.33   | 46.67 | 66.67 | 66.67 | 8.59  | 5.84  | 4.78  | 20.00  | 43.33 | 76.67 | 76.67 | 11.06 | 9.18  | 4.48  | 34.57  |
>
>
> Experimental results indicate that increasing visual input can actually degrade reasoning performance; specifically, higher frame counts do not correlate with better results in prediction tasks. The 'trajectory frame + motion description' setting achieved overall performance consistent with the 20-frame baseline, observing slight improvements in non-predictive tasks but minor declines in predictive ones. Overall, these sensitivity experiments confirm that the frame count is not the limiting factor on GTR-Bench; rather, the primary bottleneck is the models' inherent deficiency in geo-temporal reasoning capabilities.

---

> ### Author Response · Authors · 2025-11-26
> **Sincerely Look Forward to Your Feedback!**
>
> As the Rebuttal deadline approaches, we kindly hope to receive your response. Your feedback is invaluable in helping us improve this work, and we truly appreciate your time and consideration.

---

> ### Author Response · Authors · 2025-11-28
> **Gentle Follow-Up: Sincerely Look Forward to Your Feedback again!**
>
> Dear Reviewers,
>
> We hope this message finds you well. As the rebuttal deadline draws near, we kindly follow up to request your valuable feedback on our submission. Your insights are crucial for us to refine the work, and we would greatly appreciate your response at your earliest convenience.
>
> Thank you for your time and consideration!
>
> Sincerely,
>
> The Authors

---

### Official Review · Reviewer_R6Jx · 2025-10-28

**Soundness:** 3
**Presentation:** 3
**Contribution:** 2
**Rating:** 4
**Confidence:** 3

**Summary:**

This paper proposes the Geo-Temporal Reasoning Benchmark (GTR-Bench) to evaluate the geographic spatio-temporal reasoning capabilities of vision-language models in large-scale camera networks containing maps and multi-view videos. Experimental results show that current mainstream models are still insufficient for this complex task. Even the most powerful Gemini-2.5-Pro ​​achieves only approximately 34.9% accuracy, significantly lower than the human-level performance (78.61%). Analysis indicates that the problems arise from the uneven utilization of spatiotemporal context, weak temporal prediction, and difficulty aligning map data with multi-view video input.

**Strengths:**

- GTR-Bench presents a cutting-edge challenge (geo-temporal reasoning) that combines maps and multi-view videos, filling a gap in existing space-time benchmarks. This capability is crucial for real-world applications such as traffic management and emergency response.

- The research provides a reproducible benchmark, data generation, and evaluation pipeline, possessing the advantage of Scientific Reproducibility, which helps promote comparative studies within the field.

- The experiments are relatively sufficient, utilizing real-world Outdoor/Indoor data, and testing current mainstream models, which allows for a relatively comprehensive evaluation of the model's spatio-temporal reasoning capabilities.

**Weaknesses:**

- The scenario coverage is not extensive enough: The scenarios are relatively single, and the camera configurations are also quite fixed.

- Heavy Reliance on Forecasting Tasks: Among the 7 tasks in the benchmark, 4 combined tasks (NSF, TF, MTTF, CR) are strongly related to trajectory prediction and reordering, and the prediction tasks use strict ST-IoU evaluation. Although these tasks are important, GTR could also include a wider range of geographical reasoning, such as route planning/validation based on geographical knowledge, or counterfactual reasoning, to enhance the benchmark's comprehens.

**Questions:**

1. For the 7 different types of tasks, I am somewhat confused about whether the answers for different task types are multiple-choice questions (MCQ) or fill-in-the-blank questions.

2. Based on the above question, what criteria did the authors use to divide the evaluation metrics, and why are some tasks evaluated using MCQ Acc while others use ST-IoU?

3. In the main paper, the authors consistently claim "joint reasoning across multiple videos with non-overlapping fields of view," but in L741-742, they mention "especially in scenarios involving multiple cameras with overlapping coverage." Could the authors clarify this?

4. For prediction tasks like NSF, TF, and MTTF, the performance gap between models on spatial and spatio-temporal reasoning is very obvious. Could an improved evaluation method be considered? For example, for cases where the predicted camera location is correct, could the average Time IoU be reported separately to more clearly isolate and quantify the model's performance on temporal prediction?

5. Case analysis of Map-Video Alignment Errors: Topology Error and FoV Alignment Error are the main error types. Could more qualitative examples and in-depth analysis of FoV Alignment Error be provided? For example, did the model confuse the FoVs of adjacent cameras, or was it completely unable to relate elements in the image to locations on the map?

**Details Of Ethics Concerns:**

This dataset is derived from real-world scenarios and may contain human faces. Please ensure that privacy protection measures have been implemented on all facial data to mitigate privacy risks.

---

> ### Author Response · Authors · 2025-11-23
> **Response to Reviewer R6Jx(1/4)**
>
> Dear Reviewer R6Jx,
>
> We would like to thank you for providing valuable comments, and we answer the concerns raised below.
>
> > **W1**: The scenario coverage is not extensive enough: The scenarios are relatively single, and the camera configurations are also quite fixed.
>
> **1.Rationale for Environment Selection.**
>
> We argue that we weigh the diversity in the spatial-temporal complexity, the motion patterns and categories of the objects more than that in background scenes in this work, since the geo-temporal reasoning capabilities of VLMs relate more closely to the spatial-temporal cues. Adjustable camera configurations would largely enrich the scenarios, and we would be happy to take some investigation within simulated enviroment in furture work. As below, we detailly demonstrate the diversity in the spatial-temporal complexity, the motion patterns and categories of the subjects in our dataset.
>
> **2.Question Diversity.**
>
> We developed GTR-Bench to encompass tasks across varying levels of spatial-temporal complexity. We categorized these tasks based on trajectory length ($track_d$) and duration ($track_t$); a task is classified into a higher complexity tier if it exceeds the defined threshold for either metric. The classification criteria are as follows:
>
> | Category | Outdoor Criteria | Indoor Criteria |
> | :--- | :--- | :--- |
> | **Long Geo-temporal** | $track_d > 250m$ or $track_t > 30s$ | $track_d > 35m$ or $track_t > 120s$ |
> | **Medium Geo-temporal**| $150m < track_d \leq 250m$ or $15s < track_t \leq 30s$ | $25m < track_d \leq 35m$ or $60s < track_t \leq 120s$ |
> | **Short Geo-temporal** | $track_d \leq 150m$ and $track_t \leq 15s$ | $track_d \leq 25m$ and $track_t \leq 60s$ |
>
> For spatial complexity, 250m represents the scale of traversing multiple city blocks or distinct topological changes in outdoor scenarios; 35m denotes indoor multi-room or cross-floor trajectory. For temporal complexity: 30s and 120s correspond to the traversal time for vehicles and pedestrians (respectively) at average speeds, or serve as indicators for highly complex kinematic patterns.
>
> The distribution of tasks across these categories is summarized below:
>
>
> | Category | Outdoor | Indoor | Total |
> | :--- | :--- | :--- | :--- |
> | **Long Geo-temporal** | 74 | 62 | 136 |
> | **Medium Geo-temporal**| 82 | 82 | 164 |
> | **Short Geo-temporal** | 54 | 66 | 120 |
>
> Beyond spatial-temporal complexity, the benchmark ensures diversity in object and motion patterns:
> *   **Object Categories:**
>     *   **Outdoor:** Includes a variety of vehicle types (sedans, trucks, SUVs) with diverse colors.
>     *   **Indoor:** Features pedestrians of different genders, ages, and heights, with varied occupations and clothing styles.
> *   **Motion Patterns:**
>     *   **Outdoor:** Covers scenarios such as straight-line driving, intersections (crossroads, T-junctions), varying traffic flow densities, and changes in traffic light timings.
>     *   **Indoor:** Includes complex patterns like moving from indoors to outdoors, ascending/descending stairs, navigating multi-room environments, and interactions with other people or objects in the scene.
>
> So, we believe that GTR-Bench, as the first benchmark focusing on geo-temporal reasoning, shows adequate covarage of spatial-temporal complexity, diversity in objects and motion patterns. We again thank the reviewer for pointing out the scenario covarage issue, and would like to further improve our GTR-Bench series in our future works.

---

> ### Author Response · Authors · 2025-11-23
> **Response to Reviewer R6Jx(2/4)**
>
> > **W2**: Heavy Reliance on Forecasting Tasks: Among the 7 tasks in the benchmark, 4 combined tasks (NSF, TF, MTTF, CR) are strongly related to trajectory prediction and reordering, and the prediction tasks use strict ST-IoU evaluation. Although these tasks are important, GTR could also include a wider range of geographical reasoning, such as route planning/validation based on geographical knowledge, or counterfactual reasoning, to enhance the benchmark's comprehens.
>
>
> Thank you for the constructive suggestion. We agree that tasks like route planning/validation based on geographical knowledge, and counterfactual reasoning, are comprehensive tasks for evaluating MLLMs' spatial reasoning capability without considering the temporal dimension, while our work primarily focusing on geo-temporal reasoning, which not only considers geo-spatial factors, but also temporal factors. So, our tasks are strictly designed around geo-temporal reasoning, requiring multiple perspective switches between maps and videos in a camera network, joint reasoning across multiple videos, with the requirements of understanding temporal factor. Therefore, outstanding performance on our benchmark necessitates that a model reasons with three fundamental types of context:
>
> + **Geographical Context**: This includes understanding the topology of road networks, the semantic functions of POIs, and the affordances of different geographical regions.
> + **Temporal Context**: This involves comprehending the relative relationships between timestamps and the progression of an object's state.
> + **Motion Context**: This pertains to the physics of movement, including distance, direction, and speed.
>
> While the task types you mentioned are addressed in benchmarks such as RoadMap [1], USTBench [2], and CityBench [3], our context and task designs are distinct. GTR-Bench fills a critical gap left by these works, and together they constitute a multi-dimensional evaluation framework for urban scenarios. We view the current version of GTR as a solid foundation.
>
> [1]Feng S, et al. Can MLLMs Guide Me Home? A Benchmark Study on Fine-Grained Visual Reasoning from Transit Maps.
>
> [2]Lai S, et al. USTBench: Benchmarking and Dissecting Spatiotemporal Reasoning of LLMs as Urban Agents.
>
> [3]Feng J, et al. Citybench: Evaluating the capabilities of large language models for urban tasks.
>
> > **Q1**: For the 7 different types of tasks, I am somewhat confused about whether the answers for different task types are multiple-choice questions (MCQ) or fill-in-the-blank questions.
>
> > **Q2**:Based on the above question, what criteria did the authors use to divide the evaluation metrics, and why are some tasks evaluated using MCQ Acc while others use ST-IoU?
>
> **1.Why different metrics of questions.**
>
> Candidate answers for all tasks are designed as multiple-choice questions (MCQs), but with a extention time-interval fill-in-the-blank for forecasting tasks(NSF,TF and MTTF). So we have MCQ Acc in evaluation. Furthermore, to precisely measure the concrete spatio-temporal deviation from the groundtruth for forecasting task, we extract the geo-locations and time information and calculate ST-IoU with groundtruth. That's how we utilize two metrics in evaluation.
>
> **2.Metrics Division.**
>
> The selection of evaluation metrics is directly tied to the distinct dimensions of these spatio-temporal forecasting tasks:
>
> + **MCQ Acc**: This metric  evaluates the accuracy of the predicted spatial location (the camera ID). We frame this portion of the task as a multiple-choice question to isolate and measure the model's spatial reasoning ability.
>
> + **ST-IoU**: This is a more stringent metric that assesses the spatio-temporal forecasting. It calculates the Intersection over Union for both the predicted camera ID and the time interval, demanding high precision in both dimensions.
>
> To enhance clarity, we have revised Table 5 in the updated manuscript to present these two metrics separately and have removed the confusing parenthetical notation.

---

> ### Author Response · Authors · 2025-11-23
> **Response to Reviewer R6Jx(3/4)**
>
> > **Q3**: In the main paper, the authors consistently claim "joint reasoning across multiple videos with non-overlapping fields of view," but in L741-742, they mention "especially in scenarios involving multiple cameras with overlapping coverage." Could the authors clarify this?
>
>
> Sorry for the confusion. We clarify here that most of the tasks in GTR-Bench indeed requires "joint reasoning across multiple videos with non-overlapping fields of view", but a small part of the tasks (31 out of 420, approx. 7%) involves overlapping camera views, specifically in Indoor subset. These instances involve three specific camera combinations: (Cam 16, Cam 14 & Cam 09), (Cam 01 & Cam 05), and (Cam 02 & Cam 04). As these statistics demonstrate, the inclusion of these few cases does not compromise the benchmark's core focus on non-overlapping challenges; rather, they offer valuable insights into emerging issues, such as "FoV Misinterpretation" in VLMs.
>
> To avoid future confusion, we will revise the manuscript to explicitly state that overlapping scenarios are rare exceptions restricted to specific indoor settings.
>
>
>
> > **Q4**: For prediction tasks like NSF, TF, and MTTF, the performance gap between models on spatial and spatio-temporal reasoning is very obvious. Could an improved evaluation method be considered? For example, for cases where the predicted camera location is correct, could the average Time IoU be reported separately to more clearly isolate and quantify the model's performance on temporal prediction?
>
>
> Thank you for the suggestion. To better evaluate the partial spatial correctness of MLLMs on GTR-Bench, we have conducted a new analysis. For the NSF, TF, and MTTF tasks, we filtered for instances where the model correctly predicted the spatial location (camera ID) and then calculated the average ST-IoU for only these successful cases. The results are presented below:
>
> | ****/**** | **outdoor** | **outdoor** | **outdoor** | **outdoor** | **outdoor** | **outdoor** | **indoor** | **indoor** | **indoor** | **indoor** | **indoor** | **indoor** |
> | --- | --- | --- | --- | --- | --- | --- | --- | --- | --- | --- | --- | --- |
> | **name** | NSF | NSF | TF | TF | MTTF | MTTF | NSF | NSF | TF | TF | MTTF | MTTF |
> | **/** | ST-IoU |  ST-IoU (Spatially Correct) | ST-IoU | ST-IoU (Spatially Correct) | ST-IoU | ST-IoU (Spatially Correct) | ST-IoU | ST-IoU (Spatially Correct) | ST-IoU | ST-IoU (Spatially Correct) | ST-IoU | ST-IoU (Spatially Correct) |
> | **Qwen2-VL-7B-Instruct** | 5.78 | 13.33 | 0 | 0 | 10.01 | 19.36 | 3.62 | 13.59 | 0 | 0 | 0 | 0 |
> | **Qwen2-VL-2B-Instruct** | 0 | 0 | 0.28 | 0 | 0 | 0 | 0 | 0 | 0.21 | 3.09 | 0 | 0 |
> | **Qwen2.5-VL-7B-Instruct** | 0 | 0 | 0 | 0 | 0.51 | 3.09 | 0 | 0 | 0 | 0 | 0 | 0 |
> | **Qwen2.5-VL-32B-Instruct** | 0.65 | 1.63 | 0 | 0 | 15.72 | 31.43 | 3.33 | 33.33 | 0 | 0 | 0 | 0 |
> | **InternVL3-8B** | 0 | 0 | 4.79 | 0 | 5.42 | 14.79 | 0 | 0 | 0.79 | 0 | 1.67 | 25.05 |
> | **InternVL3-38B** | 8.27 | 35.46 | 8.20 | 52.45 | 20.58 | 41.17 | 11.10 | 47.59 | 4.37 | 0 | 10.24 | 61.45 |
> | **InternVL3-2B** | 6.15 | 18.46 | 0 | 0 | 9.95 | 36.07 | 0.65 | 9.78 | 0.08 | 0 | 1.62 | 45.30 |
> | **gpt-5** | 12.04 | 14.21 | 12.12 | 0 | 7.34 | 15.40 | 11.34 | 27.58 | 2.55 | 0 | 1.75 | 15.30 |
> | **gpt-4o** | 20.53 | 38.49 | 0 | 0 | 23.10 | 30.13 | 13.00 | 43.32 | 0 | 0 | 2.79 | 20.94 |
> | **GLM-4.1V-9B-Thinking** | 10.29 | 27.44 | 0 | 2.72 | 25.38 | 33.11 | 2.87 | 27.74 | 0 | 0 | 1.67 | 25.05 |
> | **gemini-2.5-pro** | 19.13 | 49.75 | 13.16 | 8.42 | 19.18 | 37.07 | 25.11 | 57.94 | 28.09 | 46.87 | 14.37 | 39.19 |
> | **claude-sonnet-4** | 8.05 | 34.12 | 6.18 | 0 | 16.94 | 25.67 | 2.60 | 9.00 | 4.01 | 17.68 | 0 | 0 |
> | **claude-3-7-sonnet-latest** | 25.75 | 70.23 | 8.90 | 24.31 | 21.97 | 29.96 | 9.48 | 40.62 | 9.51 | 49.45 | 3.56 | 53.35 |
>
> The results clearly demonstrate substantial improvements for most models under the ST-IoU (Spatially Correct) metric. This indicates that when spatial predictions are accurate, the models' temporal prediction capabilities also show corresponding growth. Concurrently, we evaluated the ST-IoU (soft) metric using relaxed spatial criteria. Experimental results indicate that all models achieved improved scores on this metric, with the TF task showing particularly substantial gains. We compared the model rankings between ST-IoU and ST-IoU (soft) and found the overall hierarchy remained largely consistent, with only minor fluctuations. This stability confirms that GTR-Bench effectively benchmarks the Geo-Temporal Reasoning capabilities across different models.

---

> ### Author Response · Authors · 2025-11-23
> **Response to Reviewer R6Jx(4/4)**
>
> > **Q5**: Case analysis of Map-Video Alignment Errors: Topology Error and FoV Alignment Error are the main error types. Could more qualitative examples and in-depth analysis of FoV Alignment Error be provided? For example, did the model confuse the FoVs of adjacent cameras, or was it completely unable to relate elements in the image to locations on the map?
>
>
> Thank you for this constructive suggestion. It‘s indeed challenging  for MLLMs to understand FoVs of adjacent cameras, or relate elements across image and map. We give more analysis on FoV Alignment Error in the following.
>
> In our benchmark, the Field of View (FoV) takes two forms: one is the **Abstract FoV**, represented by blue regions on the map; the other is the scene directly captured by the camera, which we term the **concrete FoV**. Based on this distinction, there are two types of FoV Alignment Errors:
>
> *   **FoV Misinterpretation:** This corresponds to the Reviewer’s mention of "the FoVs of adjacent cameras." It represents the model's failure to process the Abstract FoV on the map, particularly regarding adjacent cameras, as the overlapping FoVs in such scenarios make interpretation more difficult for the model. The example of "FoV Alignment Error" presented in Figure 6(b) of the main text illustrates this specific error. For more detailed examples, please refer to **Figure 11: FoV Misinterpretation** in the Appendix.
> *   **FoV to Map Misalignment:** This corresponds to the comment "relate elements in the image to locations on the map." This is a conventional error where the model fails to localize items observed in the concrete FoV onto the map. Supplementary examples are provided in **Figure 10: FoV to Map Misalignment** in the Appendix.
>
> We hope these additional examples effectively address your concerns.

---

> > ### Comment · Reviewer_R6Jx · 2025-11-28
> >
> > Most of my concerns have been addressed by the authors. I acknowledge that focusing VLM research on the Geo-temporal reasoning holds significant meaning and value. However, I still believe that the diversity of scenarios covered in the current dataset is not rich enough. Given that the Geo-temporal attribute has the potential to include a wider and more diverse range of testing scenarios, the limitations of the current dataset make me remain reserved about the paper's overall impact and generalizability. Therefore, I am willing to assign a score of 5 (if possible), but I am currently not inclined to raise it to 6.

---

> ### Author Response · Authors · 2025-11-26
> **Sincerely Look Forward to Your Feedback!**
>
> As the Rebuttal deadline approaches, we kindly hope to receive your response. Your feedback is invaluable in helping us improve this work, and we truly appreciate your time and consideration.

---

> ### Author Response · Authors · 2025-11-28
> **Gentle Follow-Up: Sincerely Look Forward to Your Feedback again!**
>
> Dear Reviewers,
>
> We hope this message finds you well. As the rebuttal deadline draws near, we kindly follow up to request your valuable feedback on our submission. Your insights are crucial for us to refine the work, and we would greatly appreciate your response at your earliest convenience.
>
> Thank you for your time and consideration!
>
> Sincerely,
>
> The Authors

---

### Official Review · Reviewer_tV6o · 2025-10-31

**Soundness:** 2
**Presentation:** 3
**Contribution:** 2
**Rating:** 4
**Confidence:** 4

**Summary:**

This paper presents GTR-Bench, a benchmark designed to evaluate models on geo-temporal reasoning, which requires understanding spatial, temporal, and motion relationships across multiple camera views and corresponding maps. It defines seven tasks spanning basic perception and higher-level reasoning, using diverse real-world indoor and outdoor video data. The benchmark contains 420 annotated examples drawn from 364 video clips and introduces a new metric, Spatio-Temporal Intersection over Union, to jointly assess spatial and temporal accuracy. Evaluations of twelve vision–language–action models show a wide gap between human and model performance, with the best system achieving only 34.9 percent accuracy compared to 78.6 percent for human participants. The results highlight persistent weaknesses in temporal reasoning, spatial alignment, and integration of map-based context. GTR-Bench establishes a standardized and challenging platform for advancing compositional reasoning and embodied intelligence in multi-view visual understanding.

**Strengths:**

1. Defining a new reasoning paradigm—geo-temporal reasoning—that fuses spatial, temporal, and motion inference across non-overlapping camera views. The task design moves beyond egocentric video QA benchmarks.
2. The benchmark construction is methodical and transparent, featuring rigorous trajectory calibration, LLM-assisted task generation, and human validation. The ST-IoU metric elegantly integrates spatial and temporal correctness.
3. By quantifying the gap between human and model reasoning, GTR-Bench establishes a standardized, scalable platform likely to become a reference benchmark for spatiotemporal multimodal reasoning.

**Weaknesses:**

1. The 20-frame input cap may underrepresent temporal dynamics, biasing against models with differing context lengths. Frame-sensitivity analysis is absent.
2. LLM-generated distractors may embed linguistic artifacts exploitable by text-heavy models; indoor/outdoor scene imbalance further limits transferability.
3. Calibration relies on manual homographies and lacks quantified reprojection or timing errors, which could affect motion and interval labels.

**Questions:**

1. How sensitive are results to the frame sampling budget (e.g., 20 vs. 40 frames)?
2. What are the quantitative calibration errors (pixel–meter, time sync) and their propagation to label uncertainty?
3. Could a soft ST-IoU variant (e.g., adjacency-aware) better reflect partial spatial correctness?
4. How were LLM-generated distractors validated for semantic plausibility and difficulty?
5. What was the human evaluation protocol—annotator expertise, inter-rater agreement, and consistency with model constraints?

---

> ### Author Response · Authors · 2025-11-23
> **Response to Reviewer tV6o(1/5)**
>
> Dear Reviewer tV6o,
>
> We sincerely thank you for this insightful question! We reply to this question as follows:
>
> > **W1**: The 20-frame input cap may underrepresent temporal dynamics, biasing against models with differing context lengths. Frame-sensitivity analysis is absent.
>
> > **Q1**: How sensitive are results to the frame sampling budget (e.g., 20 vs. 40 frames)?
>
> **1.Why we limit to 20 frames.**
>
> Our experimental design restricts the total video input to a cumulative 20 frames. This constraint is necessary to accommodate the high-resolution imagery required for motion state recognition within the context window limits of specific models. This design ensures that our results primarily reflect the model's ability to reason about geo-temporal relationships, rather than its proficiency in decoding long-form temporal dynamics.
>
> **2.Frame-sensitivity analysis on GTR-Bench.**
>
> To evaluate sensitivity to frame sampling budgets, we conducted a new series of experiments for Gemini-2.5-pro with three distinct settings:
> + **20-frame limit**: The baseline setting used in our original paper.
> + **32-frame limit**: We increased the sampling budget to 32 frames.
> + **Trajectory frame + motion description**: To decouple the reasoning component from temporal perception, we provided a single composite image displaying the target's full trajectory, augmented with a textual description derived from ground-truth motion data.
>
> The results are presented in the table below.
>
>   |                                       | outdoor |       |       |       |       |       |       | indoor |       |       |       |       |       |       | avg    |
> |---------------------------------------|---------|-------|-------|-------|-------|-------|-------|--------|-------|-------|-------|-------|-------|-------|--------|
> | video type                            | ATI     | MS    | GL    | CR    | NCF   | TF    | MTTF  | ATI    | MS    | GL    | CR    | NCF   | TF    | MTTF  | avg    |
> | 20-frame limit                        | 60.00   | 46.67 | 33.33 | 56.67 | 19.13 | 13.16 | 19.18 | 63.33  | 13.33 | 26.67 | 70.00 | 25.11 | 28.09 | 14.37 | 34.93  |
> | 32-frame limit                        | 36.67   | 50.00 | 46.67 | 26.67 | 16.25 | 6.11  | 4.72  | 16.67  | 30.00 | 56.67 | 70.00 | 19.92 | 11.35 | 12.41 | 28.87  |
> | trajectory frame + motion description | 43.33   | 46.67 | 66.67 | 66.67 | 8.59  | 5.84  | 4.78  | 20.00  | 43.33 | 76.67 | 76.67 | 11.06 | 9.18  | 4.48  | 34.57  |
>
> Experimental results indicate that increasing visual input can actually degrade reasoning performance; specifically, higher frame counts do not correlate with better results in prediction tasks. The 'trajectory frame + motion description' setting achieved overall performance consistent with the 20-frame baseline, observing slight improvements in non-predictive tasks but minor declines in predictive ones. Overall, these sensitivity experiments confirm that the frame count is not the limiting factor on GTR-Bench; rather, the primary bottleneck is the models' inherent deficiency in geo-temporal reasoning capabilities.

---

> ### Author Response · Authors · 2025-11-23
> **Response to Reviewer tV6o(2/5)**
>
> > **W2**: LLM-generated distractors may embed linguistic artifacts exploitable by text-heavy models; indoor/outdoor scene imbalance further limits transferability.
>
> > **Q4**: How were LLM-generated distractors validated for semantic plausibility and difficulty?
>
> **1. No LLM, but rule-based distractor construction.**
>
> We clarify that our question construction is involved a rule-based approach followed by manual refinement, without LLMs for generating distractors. Instead, we designed an option construction framework that is rule-based, logically rigorous, and supplemented with manual polishing.
>
> **2. Rule-based Difficulty Control.**
>
> We have meticulously designed the generation strategies for the three core elements of distractors: Time Interval, Geographic Location, and Motion State.
>
> The time interval for the distractor is then defined as [$gt_{low} - k_{low}$, $gt_{up} + k_{up}$]. By adjusting the range of $k\sim \mathcal{N}(\mu, \sigma^2)$, we can control the difficulty of the distractors for indoor and outdoor scenes.
>
> For geographic location (camera ID), we select camera IDs that are within the minimum enclosing circle radius $r$ of the target's true trajectory but are not the ground-truth. We prioritize using real cameras that have associated video data to construct the options. In specific situations (especially with sparse camera distribution), if suitable real cameras cannot be found for distractor options, we will generate "fake" camera IDs based on the road network.
>
> When comes to Motion state, We only replace any one of the three indicators direction, distance, and speed in motion discreption. It is worth noting that to further enhance the evaluation dimensions of the model, we have modified and polished the sentence structure and expression of the incorrect and distractor options to make the forms of the options more rich and diverse.

---

> ### Author Response · Authors · 2025-11-23
> **Response to Reviewer tV6o(3/5)**
>
> > **W3**: Calibration relies on manual homographies and lacks quantified reprojection or timing errors, which could affect motion and interval labels.
> >
> > **Q2**: What are the quantitative calibration errors (pixel–meter, time sync) and their propagation to label uncertainty?
>
> **1.Calibration data source**
>
> Thank you for your our work builds upon two well-established multi-camera ReID datasets, CityFlow [1] and MTMMC [2]. We are indebted to the creators of these datasets for providing accurately synchronized and timestamped bounding box sequences, featuring professionally aligned temporal data. The CityFlow dataset already includes a complete and accurate set of homography matrices. Therefore, we utilized these pre-existing calibrations directly. Our primary annotation effort was consequently focused on generating high-quality homography matrices for MTMMC.
>
> **2.Controllable error**
>
> Besides, we conducted a thorough reprojection error analysis for every camera, as shown in the table below.
>
> | Scene (Dataset) | Statistic | Reprojection Error (pixels) | Reprojection Error (meters) |
> | :--- | :--- | :--- | :--- |
> | **Outdoor (CityFlow)** | Average | 13.25 | 2.18 |
> | | Minimum | 2.19 (S04_c039) | 0.20 (S04_c031) |
> | | Maximum | 21.26 (S04_c024) | 14.52 (S04_c029) |
> | **Indoor (MTMMC)** | Average | 12.83 | 1.18 |
> | | Minimum | 1.37 (c04) | 0.12 (c04) |
> | | Maximum | 35.06 (c16) | 3.12 (c16) |
>
> The average metric error in the outdoor scenes is 2.18 meters. When considered in the context of the average inter-camera distance of 984.77 meters, this error is controlled to within 0.22%. For the indoor scenes, the average metric error is 1.18 meters. Given the average inter-camera distance of 30.59 meters, this error is maintained within 4%. Furthermore, the spatial perturbation scale of our designed distractors exceeds 5m for indoor settings and 15m for outdoor settings. Consequently, these margins of error do not impact the validity of our conclusions.
>
> Some cameras will inevitably exhibit higher reprojection errors, such as camera S04_c024 in CityFlow and c16 in MTMMC. To mitigate the propagation of these errors, we have instituted a stage of manual verification to ensure the overall accuracy and integrity of GTR-Bench.
>
> [1]Tang Z, et al. Cityflow: A city-scale benchmark for multi-target multi-camera vehicle tracking and re-identification.
>
> [2]Woo S, et al. MTMMC: A Large-Scale Real-World Multi-Modal Camera Tracking Benchmark.

---

> ### Author Response · Authors · 2025-11-23
> **Response to Reviewer tV6o(4/5)**
>
> > **Q3**: Could a soft ST-IoU variant (e.g., adjacency-aware) better reflect partial spatial correctness?
>
>
> We introduced a "soft" version of the ST-IoU metric to better assess the models' understanding of spatial relationships. Our soft ST-IoU employs a stepped scoring mechanism. Specifically, we calculate the distance, $d$, between each predicted camera and the ground-truth camera. A base distance, termed $scale$, is established for indoor(5m) and outdoor(15m). Scores are allocated based on this distance, in the following table:
>
> | Distance (d) | Soft Score |
> | :--- | :--- |
> | $d < scale$ | 1 |
> | $scale <= d < 2*scale$ | 0.5 |
> | $2*scale <= d < 3*scale$ | 0.2 |
> | $d >= 3*scale$ | 0 |
>
> The formula for the Soft ST-IoU is defined as:
>
> $$\text{ST-IoU(Soft)} = \frac{1}{N} \sum_{i=1}^{N} \text(Soft Score(C_{p_i},C_{gt_i})) \times \frac{|T_{p_i} \cap T_{gt_i}|}{|T_{p_i} \cup T_{gt_i}|}$$
>
> Our experimental results are shown below.
>
>
> | / | GTR-Outdoor |  |  |  |  |  | GTR-Indoor |  |  |  |  |  |  |  |  |  |
> | --- | --- | --- | --- | --- | --- | --- | --- | --- | --- | --- | --- | --- | --- | --- | --- | --- |
> | name | NSF | NSF | TF | TF | MTTF | MTTF | NSF | NSF | TF | TF | MTTF | MTTF | avg | avg | rank | rank |
> | / | ST-IoU | ST-IoU (Soft) | ST-IoU | ST-IoU (Soft) | ST-IoU | ST-IoU (Soft) | ST-IoU | ST-IoU (Soft) | ST-IoU | ST-IoU (Soft) | ST-IoU | ST-IoU (Soft) | ST-IoU | ST-IoU (Soft) | ST-IoU | ST-IoU (Soft) |
> | gemini-2.5-pro | 19.13 | 21.87 | 13.16 | 15.83 | 19.18 | 19.18 | 25.11 | 29.71 | 28.09 | 30.73 | 14.37 | 20.74 | 19.84 | 23.01 | 1 | 1 |
> | claude-3-7-sonnet-latest | 25.75 | 26.37 | 8.90 | 26.79 | 21.97 | 21.97 | 9.48 | 18.53 | 9.51 | 21.54 | 3.56 | 5.84 | 13.19 | 20.17 | 2 | 2 |
> | InternVL3-38B | 8.27 | 8.27 | 8.20 | 18.51 | 20.58 | 20.58 | 11.10 | 12.95 | 4.37 | 11.54 | 10.24 | 11.93 | 10.46 | 13.97 | 3 | 4 |
> | gpt-4o | 20.53 | 21.76 | 0 | 18.79 | 23.10 | 23.10 | 13.00 | 16.65 | 0 | 16.40 | 2.79 | 4.58 | 9.90 | 16.88 | 4 | 3 |
> | gpt-5 | 12.04 | 14.97 | 12.12 | 13.55 | 7.34 | 7.92 | 11.34 | 10.62 | 2.55 | 10.22 | 1.75 | 4.90 | 7.86 | 10.36 | 5 | 8 |
> | GLM-4.1V-9B-Thinking | 10.29 | 10.83 | 0 | 14.83 | 25.38 | 25.38 | 2.87 | 8.45 | 0 | 13.44 | 1.67 | 2.00 | 6.70 | 12.49 | 6 | 5 |
> | claude-sonnet-4 | 8.05 | 11.46 | 6.18 | 15.95 | 16.94 | 18.21 | 2.60 | 6.11 | 4.01 | 16.41 | 0 | 2.09 | 6.30 | 11.71 | 7 | 6 |
> | Qwen2.5-VL-32B-Instruct | 0.65 | 12.55 | 0 | 20.52 | 15.72 | 19.84 | 3.33 | 7.11 | 0 | 7.00 | 0 | 2.87 | 3.28 | 11.65 | 8 | 7 |
> | Qwen2-VL-7B-Instruct | 5.78 | 9.11 | 0 | 6.83 | 10.01 | 10.77 | 3.62 | 9.88 | 0 | 2.00 | 0 | 1.11 | 3.24 | 6.62 | 9 | 10 |
> | InternVL3-2B | 6.15 | 6.15 | 0 | 11.50 | 9.95 | 9.95 | 0.65 | 6.15 | 0.08 | 2.34 | 1.62 | 2.68 | 3.08 | 6.46 | 10 | 11 |
> | InternVL3-8B | 0 | 0 | 4.79 | 8.10 | 5.42 | 5.42 | 0 | 0 | 0.79 | 4.52 | 1.67 | 1.67 | 2.11 | 3.29 | 11 | 13 |
> | Qwen2.5-VL-7B-Instruct | 0 | 9.95 | 0 | 10.86 | 0.51 | 2.92 | 0 | 2.14 | 0 | 1.07 | 0 | 0.31 | 0.09 | 4.54 | 12 | 12 |
> | Qwen2-VL-2B-Instruct | 0 | 15.70 | 0.28 | 6.48 | 0 | 15.71 | 0 | 2.35 | 0.21 | 0.21 | 0 | 0.34 | 0.08 | 6.80 | 13 | 9 |
>
>
> Experimental results indicate that all models achieved improved scores on the ST-IoU (soft) metric, with the TF task showing particularly substantial gains. We compared the model rankings between ST-IoU and ST-IoU (soft) and found the overall hierarchy remained largely consistent, with only minor fluctuations. This stability confirms that GTR-Bench effectively benchmarks the Geo-Temporal Reasoning capabilities across different models.

---

> ### Author Response · Authors · 2025-11-23
> **Response to Reviewer tV6o(5/5)**
>
> > **Q5**: What was the human evaluation protocol—annotator expertise, inter-rater agreement, and consistency with model constraints?
>
> We provide a detailed description of our human evaluation protocol below.
>
> 1. Annotator Background and Qualifications
>
> To ensure the reliability of our human evaluation results, we implemented a rigorous screening and training process for the personnel involved in data annotation and performance assessment.
>
> a. Composition of Annotators and Evaluators:
>
> + **Professional Field**: The dataset was constructed by researchers with profound academic backgrounds in 3D vision.
> + **Professional Experience**: These researchers possess extensive experience in multi-view geometry, scene understanding, and spatial intelligence.
> + **Knowledge Requirements**: All were required to have a deep understanding of core concepts such as spatial relationships, camera poses, and object motion, and be capable of complex spatial reasoning.
> + **Distinction between Annotators and Evaluators**: The personnel responsible for question annotation were a separate group from those who conducted the benchmark evaluation, ensuring the independence of results.
>
> b. Annotation Guidelines
>
> + **Multi-reviewer Mechanism**: During the dataset construction, each sample generated by our rules was cross-validated by multiple other independent annotators.
> + **Dependence on Maps and Camera Videos**: We ensured that questions could not be answered solely through the map, camera video, or common sense, but required the synthesis of information from all provided map and video inputs.
> + **Clarity and Unambiguity of Context and Questions**: We ensured that the content of the map images and camera videos was clear and that the question text was precise and unambiguous.
> + **Uniqueness of Answer**: We ensured that there was only one correct answer and that the distractors were designed to be plausible.
> + **Spatio-Temporal Span Coverage**: All seven question types in the benchmark were required to have diversity in their temporal and spatial spans to ensure the variety of the problems across different spatio-temporal dimensions.
> + **Ethics and Licensing**: We ensured that all videos and maps were either fully copyrighted or used with the author's consent, and we maintained the same strict standards for personal privacy information as the reference datasets.
>
> c. Evaluation Guidelines
>
> Human performance score was the average accuracy by independent evaluators. This multi-evaluator mechanism reduces the randomness caused by individual differences, making the evaluation results more statistically significant and stable.
>
> 2. Consistency with Model Constraints
>
> Evaluators had access to the same information as the model input. They were forbidden from using any external knowledge or tools for assistance. For full evaluation, we designed a front-end evaluation page, as shown in the appendix . This interface was built to  present all map and camera information necessary to answer the question and allow evaluators to zoom and draw on the images to mark key elements during their reasoning process. Besides, we provided human evaluators with unrestricted evaluation time to measure the optimal performance of humans under ideal conditions, revealing the gap between human and MLLMs.

---

> ### Author Response · Authors · 2025-11-26
> **Sincerely Look Forward to Your Feedback!**
>
> As the Rebuttal deadline approaches, we kindly hope to receive your response. Your feedback is invaluable in helping us improve this work, and we truly appreciate your time and consideration.

---

> ### Author Response · Authors · 2025-11-28
> **Gentle Follow-Up: Sincerely Look Forward to Your Feedback again!**
>
> Dear Reviewers,
>
> We hope this message finds you well. As the rebuttal deadline draws near, we kindly follow up to request your valuable feedback on our submission. Your insights are crucial for us to refine the work, and we would greatly appreciate your response at your earliest convenience.
>
> Thank you for your time and consideration!
>
> Sincerely,
>
> The Authors

---

### Official Review · Reviewer_EAyu · 2025-11-01

**Soundness:** 3
**Presentation:** 3
**Contribution:** 3
**Rating:** 6
**Confidence:** 3

**Summary:**

This paper proposes GTR Bench a geo-temporal benchmark that links real multi-camera videos with map context to test VLM reasoning. The paper studies the missing evaluation of VLMs on joint map plus multi camera spatial temporal reasoning unlike STI Bench or MapEval which use only videos or only maps. Experiments like Table 4 show that humans get 78.61% accuracy while popular modles like Gemini 2.5 Pro gets 34.93% showing current models fail on this benchmark.

**Strengths:**

- The reviewer finds the proposed idea to extend spatial temporal VLM benchmarks such as STI Bench for driving to true geo-temporal multi camera settings to be interesting.
- A surprising fact is that even strong proprietary models still lag humans by more than 40 points on the same tasks which clearly supports the need for this benchmark.
- Interestingly, experiments in Table 4, Table 6, and Figure 4 report 12 models across indoor and outdoor tasks and show consistent drops when map reasoning is required.
- Writing is mostly clear.

**Weaknesses:**

- The LLM based question and distractor generation in the dataset construction section does not show an ablation on how prompts preserve spatial relations across cameras and the closest related method is automatic MCQ generation in STI Bench.
- The experiments do not compare against easy multi-target multi -camera tracking or ReID baselines that already exist for CityFlow and MTMMC even though the data source is aligned so such baselines should have been trivial to include.
- Some typos: Line 179: multipe -> multiple; Line 180: resoning -> reasoning; L248: exmples -> examples; L146: geograohic -> geographic.

**Questions:**

- In Table 4 (page 8), Gemini 2.5 Pro scores 25.11 on Indoor NSF, while in Table 5 (page 9), its Indoor NSF-MCQ accuracy is 43.33. These two metrics both test indoor navigation reasoning but show very different performance. Could you clarify if these datasets overlap or if the question format accounts for this difference?

---

> ### Author Response · Authors · 2025-11-23
> **Response to Reviewer EAyu(1/4)**
>
> Dear Reviewer EAyu,
>
> We sincerely thank you for the insightful comments!
>
> > **W1**: The LLM based question and distractor generation in the dataset construction section does not show an ablation on how prompts preserve spatial relations across cameras and the closest related method is automatic MCQ generation in STI Bench.
>
> **1.How we preserve spatial relations.**
>
> In our data construction, distractor questions are produced by perturbing the temporal/geolocation ground-truth in rule-based manner followed by manul refiement, rather than generating distractor questions by LLMs in STI-Bench[1].
>
> **2.More discussion for preserving spatial relations in context.**
>
> To show the ablation on how prompts preserve spatial relations across cameras, we further introduce textual map context (visual map context was used in initial setup) for supplementary evaluations. The two approaches are detailed as follows:
>
> + **Visual Map Context**: We utilize a rendered map image from OpenStreetMap, which precisely annotates the locations, orientations, Camera ID, and FoV of all candidate cameras. Furthermore, we have explicitly included a north-arrow and a scale bar for absolute geographical orientation and scale references.
>
> + **Textual Map Context**: We have also designed a text-only prompt to describe the spatial relationships between cameras, which translates the topological relations and relative positions from the map (e.g., Camera A is located 50 meters northeast of Intersection B, facing southeast) into structured natural language descriptions.
>
> Based on these, we performed an ablation study to evaluate the impact of visual and textual context on the model's ability to preserve spatial relationships. We benchmarked two SOTA models, Gemini-2.5-pro and InternVL-38B on GTR-Bench. The results are presented below.
>
> | | | **GTR-Outdoor** | | | | | | | **GTR-Indoor** | | | | | | |
> |---|---|---|---|---|---|---|---|---|---|---|---|---|---|---|---|
> | **Model** | **Map Type** | **ATI** | **MS** | **GL** | **CR** | **NCF** | **TF** | **MTTF** | **ATI** | **MS** | **GL** | **CR** | **NCF** | **TF** | **MTTF** |
> | gemini-2.5-pro | image | 60.00 | 46.67 | 33.33 | 56.67 | 19.13 | 13.16 | 19.18 | 63.33 | 13.33 | 26.67 | 70.00 | 25.11 | 28.09 | 14.37 |
> | gemini-2.5-pro | text | 43.33 | 56.67 | 60.00 | 66.67 | 6.83 | 0.00 | 1.11 | 13.33 | 43.33 | 73.33 | 83.33 | 27.76 | 19.76 | 21.42 |
> | InternVL3-38B | image | 40.00 | 73.33 | 30.00 | 53.33 | 8.27 | 8.20 | 20.58 | 50.00 | 56.67 | 26.67 | 37.93 | 11.10 | 4.37 | 10.24 |
> | InternVL3-38B | text | 38.33 | 33.33 | 76.67 | 36.36 | 3.06 | 1.08 | 3.99 | 50.00 | 33.33 | 56.67 | 73.33 | 4.36 | 0.55 | 9.20 |
>
> The results clearly indicate that there is no single best method for encoding spatial relationships. Text-based prompts tend to excel in tasks requiring abstract or logical spatial reasoning.  Image-based prompts demonstrate a strong advantage in tasks that rely on understanding physical motion and trajectories.
>
>
> [1] Li Y, et al. Sti-bench: Are mllms ready for precise spatial-temporal world understanding?

---

> ### Author Response · Authors · 2025-11-23
> **# Response to Reviewer EAyu(2/4)**
>
> > **W2**: The experiments do not compare against easy multi-target multi -camera tracking or ReID baselines that already exist for CityFlow and MTMMC even though the data source is aligned so such baselines should have been trivial to include.
>
> **1.Unfairness evaluation between ReID baseline and MLLMs**
>
> We must clarify that ReID baselines face inherent limitations on GTR-Bench. Specifically, the benchmark restricts access to option-associated videos for all tasks, whereas standard ReID requires extracting visual content from these options, precluding a fair comparison. Despite this, we implemented ReID baselines for the GL, ATI, NSF, TF, and MTTF tasks using the SBS algorithm (an enhanced BoT [1]) from FastReid [2]. Tasks requiring complex semantic reasoning (MS and CR) were excluded as they are infeasible under this setting.
>
> **2. ReID baseline is merely comparable to the SOTA MLLMs on unfairness evaluation**
>
> For baseline, we first extract a target query image from the context video. We then apply YOLOv8 to the candidate video to generate bounding boxes, perform ReID matching, and identify time intervals of successful detections. To align options with video content: for spatial part (Camera ID), we analyze video within the specified time range and select the option with the highest similarity; for temporal part(time interval), we aggregate timestamps exceeding a similarity threshold to form time intervals. Specifically, we clip  videos in trajectory time for ATI and utilize video in future for forecasting tasks. We used class-specific weights for ReID, setting similarity thresholds at 0.4 for vehicles and 0.96 for persons. The experimental results are as follows:
>
> | Task | ReID Baseline |  | Gemini-2.5-Pro |  |
> | --- | --- | --- | --- | --- |
> |  | **MCQ Acc (%)** | **ST-IoU** | **MCQ Acc (%)** | **ST-IoU** |
> | **ATI indoor** | 16.67 | N/A | 13.33 | N/A |
> | **ATI outdoor** | 23.33 | N/A | 46.67 | N/A |
> | **GL indoor** | 46.67 | N/A | 63.33 | N/A |
> | **GL outdoor** | 63.33 | N/A | 60.00 | N/A |
> | **MTTF indoor** | 52.17 | 30.98 | 36.67 | 14.37 |
> | **MTTF outdoor** | 66.67 | 22.24 | 51.72 | 19.18 |
> | **NSF indoor** | 43.33 | 16.03 | 43.33 | 25.11 |
> | **NSF outdoor** | 66.67 | 43.53 | 38.46 | 19.13 |
> | **TF indoor** | 40.00 | 9.39 | 50.00 | 28.09 |
> | **TF outdoor** | 38.33 | 16.12 | 45.45 | 13.16 |
> | **Average** | 45.72 | 23.05 | 44.90 | 19.84 |
>
> Experimental results indicate that despite the ReID baseline having direct access to video context context, its performance is merely comparable to the state-of-the-art Gemini-2.5-Pro. This finding validates the significant Geo-Temporal reasoning capabilities of MLLMs, demonstrating their ability to leverage graphic map and video context to effectively reason about and predict unobserved events.
>
> [1] Luo H, et al. Bag of tricks and a strong baseline for deep person re-identification.
> [2] He L, et al. Fastreid: A pytorch toolbox for general instance re-identification.

---

> ### Author Response · Authors · 2025-11-23
> **Response to Reviewer EAyu(3/4)**
>
> > **W3**: Some typos: Line 179: multipe -> multiple; Line 180: resoning -> reasoning; L248: exmples -> examples; L146: geograohic -> geographic.
>
> Thank you for pointing out these typos, which have been corrected corrected all of them in the revised manuscript:
> + Line 179: "multipe" has been corrected to "multiple".
> + Line 180: "resoning" has been corrected to "reasoning".
> + Line 248: "exmples" has been corrected to "examples".
> + Line 146: "geograohic" has been corrected to "geographic".

---

> ### Author Response · Authors · 2025-11-23
> **Response to Reviewer EAyu(4/4)**
>
> > **Q1**: In Table 4 (page 8), Gemini 2.5 Pro scores 25.11 on Indoor NSF, while in Table 5 (page 9), its Indoor NSF-MCQ accuracy is 43.33. These two metrics both test indoor navigation reasoning but show very different performance. Could you clarify if these datasets overlap or if the question format accounts for this difference?
>
> To avoid confusion, we emphasize that there is only one dataset. The data points in question simply represent different evaluation metrics, not different data sources. For each question in the spatio-temporal forecasting tasks (NSF, TF, and MTTF), the model is required to predict both a spatial location (e.g., Camera ID C.C017) and a temporal interval (e.g., 12:00:37-12:00:42). To comprehensively assess the model's performance, we employ two distinct metrics:
>
> + **MCQ Acc(spatial)**: This metric isolates and evaluates only the spatial component of forecasting tasks. We frame this sub-task as a multiple-choice question where the model must select the correct camera ID from a set of options.
>
> + **ST-IoU((spatial-temporal)**: This is a rigorous metric that evaluates the entire spatio-temporal forecasting. A high score in ST-IoU demands precision in both the spatial and temporal dimensions.
>
> The observed performance difference is expected. Achieving a high MCQ accuracy by correctly identifying the spatial location is a necessary but insufficient condition for achieving a high ST-IoU score. The latter requires the model to also forecast the time interval with high fidelity.

---

> ### Author Response · Authors · 2025-11-26
> **Sincerely Look Forward to Your Feedback!**
>
> As the Rebuttal deadline approaches, we kindly hope to receive your response. Your feedback is invaluable in helping us improve this work, and we truly appreciate your time and consideration.

---

> ### Author Response · Authors · 2025-11-28
> **Gentle Follow-Up: Sincerely Look Forward to Your Feedback again!**
>
> Dear Reviewers,
>
> We hope this message finds you well. As the rebuttal deadline draws near, we kindly follow up to request your valuable feedback on our submission. Your insights are crucial for us to refine the work, and we would greatly appreciate your response at your earliest convenience.
>
> Thank you for your time and consideration!
>
> Sincerely,
>
> The Authors

---

### Author Response · Authors · 2025-11-23
**To all**

Dear AC and all reviewers:

We appreciate that reviewers recognized the following merits of our work:

- **Novelty and soundness ([EAyu, tV6o, R6Jx]):** We define a noval paradigm that fuses spatial, temporal, and motion inference across non-overlapping camera views.
- **Methodical and reproducible construction ([tV6o, R6Jx, f47F]):** The benchmark features rigorous trajectory calibration, transparent data generation, and an ST-IoU metric. Our standardized evaluation protocol and balanced task design ensure scientific reproducibility for future comparative studies.
- **Sufficient and comprehensive experiments ([EAyu, R6Jx, f47F]):** Extensive evaluations with diverse VLMs on real-world indoor and outdoor data reveal a significant gap between current models and human performance, validating the necessity and difficulty of the benchmark.

We also thank all reviewers for their insightful and constructive suggestions, which help further improve our paper. In addition to the pointwise responses below, we summarize the major revision in the rebuttal according to the reviewers' suggestions.

- **Additional experiments ([EAyu, tV6o, R6Jx, f47F])**: We add further experiments to verify our designs in three folds: (1) We benchmark traditional ReID methods and compare text vs. visual map contexts ([EAyu]). (2)The frame sensitivity analysis comparing 20/32-frame and trajectory+text validates our sampling design ([tV6o, f47F]). (3) Furthermore, we provide quantitative reprojection error checks  and a detailed qualitative categorization of Map-Video alignment errors ([tV6o，R6Jx]).

- **Manuscript clarification ([EAyu, tV6o, R6Jx, f47F])**: We explained the evaluation characteristics of different benchmark metrics and distractor generation strategies ([EAyu, tV6o, R6Jx]). We provide detailed information on complexity criteria, distribution statistics and the rarity of overlapping scenarios ([R6Jx, f47F]). In addition, we introduce a Soft ST-IoU metric to validate ranking stability  and perform a decoupled analysis to isolate temporal reasoning ([tV6o, R6Jx]). Besides, we add specific visual examples of FoV errors ([R6Jx]).

Our detailed responses can be found in corresponding comments replyig to each reviewer. We hope our responses can clarify the confusion and address the raised concerns. Again, we sincerely thank all reviewers for their efforts and time.

Best regards,

The Authors

---

### Public Comment · ~Qinghongbing_Xie1 · 2026-02-28
**Camera Ready Version**

Dear Program Chairs, Area Chair, and Reviewers,

We have uploaded the camera-ready version of GTR-Bench. We sincerely thank the committee for the positive recommendations and the insightful feedback provided throughout the review process.

To address the constructive suggestions from the review and discussion phases, we have integrated the following major updates into the final manuscript:

**Dataset Statistics and Complexity(Section.4):** We provided comprehensive details regarding our complexity criteria, dataset distribution statistics

**Additional Baselines and Analysis （Section.5 and Appendix) :** We have included benchmark results for traditional ReID methods, compared text versus visual map contexts to fully validate our sampling design.

**Error Verification and Categorization(Appendix):** We incorporated quantitative reprojection error checks, alongside a detailed qualitative categorization of Map-Video alignment errors and specific visual examples of FoV errors.

**Soft ST-IoU and Analysis(Appendix):** We introduced a new Soft ST-IoU metric to validate ranking stability and performed a decoupled analysis to explicitly isolate temporal reasoning.


We are grateful for the guidance that helped strengthen the final manuscript and look forward to presenting our work.

Sincerely,

The Authors

---

### Meta-Review · Area_Chair_m6cZ · 2026-01-06

**Summary:**

The reviewers generally agree that the paper addresses a significant and timely gap in the evaluation of Vision-Language Models (VLMs): the integration of geographic perspective (maps) with multi-camera video context. The reviewers praised the novelty of the Geo-Temporal Reasoning (GTR) paradigm, the methodical construction of the benchmark, and the introduction of the ST-IoU metric.
The primary concerns that informed the discussion were:

- Diversity and Coverage: Multiple reviewers expressed concerns regarding the limited number of scenarios (two main environments) and whether these were sufficient to draw strong statistical conclusions.

- Input Constraints (Frame Sampling): Reviewers questioned whether the 20-frame input cap underrepresented temporal dynamics or disadvantaged long-context models.

- Baselines and Ablations: Reviewer requested comparisons with traditional ReID/tracking baselines and an ablation study on the impact of different prompt contexts (visual vs. textual maps).

- Metric Clarity: Reviewer requested a clearer breakdown of MCQ vs. ST-IoU metrics and a more granular analysis of temporal vs. spatial errors.

**Reviewer Concerns:**

The authors provided a very thorough rebuttal, including several new experiments and manuscript revisions that successfully addressed the majority of the technical concerns:

(At least partially) Addressed:

- Baselines & Prompts: The authors conducted a prompt ablation study and implemented ReID baselines (using the SBS algorithm), demonstrating that traditional tracking methods significantly lag behind MLLMs and human performance, validating the benchmark's difficulty.

- Frame Sensitivity: The authors performed a sensitivity analysis (Gemini-2.5-Pro on 20 vs. 32 frames). Results showed that increased frames did not correlate with better results and occasionally degraded performance, suggesting the bottleneck is reasoning, not perception.

- Distractors: The authors clarified that distractor generation was rule-based and manually refined, mitigating concerns about linguistic artifacts exploitability.

- Metric Clarity: A "Soft ST-IoU" was introduced to account for partial spatial correctness, and the authors provided a decoupled analysis of spatial vs. temporal forecasting errors.

Partially Outstanding:

Scenario Diversity: While the authors added a detailed breakdown of task complexity (Short/Medium/Long Geo-temporal) to show internal variety, I think some reviewers would still remain somewhat reserved about the limited number of real-world environments (two) for a generalizable benchmark.

**Reviewer Scores:**

- Reviewer EAyu (6): Likely to maintain or move to a 7 given the responsive inclusion of requested baselines.
- Reviewer tV6o (4): Did not officially update, but the frame-sensitivity analysis directly countered their main technical objection. Likely a shift to 5.
- Reviewer R6Jx (4 → 5): Officially increased to 5 following the rebuttal, acknowledging addressed concerns while remaining reserved on diversity.
- Reviewer f47F (4): Concerns regarding the 20-frame limit were effectively debunked by the authors' sensitivity study. While coverage concerns remain, the technical rigor of the response likely moves them to a 5.

---

### Decision · Program_Chairs · 2026-01-26

Accept (Poster)